# Human Adult Astrocyte Extracellular Vesicle Transcriptomics Study Identifies Specific RNAs Which Are Preferentially Secreted as EV Luminal Cargo

**DOI:** 10.3390/genes14040853

**Published:** 2023-03-31

**Authors:** Keerthanaa Balasubramanian Shanthi, Daniel Fischer, Abhishek Sharma, Antti Kiviniemi, Mika Kaakinen, Seppo J. Vainio, Geneviève Bart

**Affiliations:** 1FBMM, Disease Networks Research Unit, Laboratory of Developmental Biology, University of Oulu, 90014 Oulu, Finland; 2Applied Statistical Methods, Natural Resources Institute Finland (Luke), Myllytie 1, 31600 Jokioinen, Finland; 3Biocenter Oulu, University of Oulu, and Oulu Centre for Cell-Matrix Research, Faculty of Biochemistry and Molecular Medicine, 90014 Oulu, Finland; 4Kvantum Institute, Infotech Oulu, University of Oulu, 90014 Oulu, Finland

**Keywords:** human adult astrocyte, extracellular vesicles, miRNA, mRNA, unannotated RNA, miRNA-seq, mRNA-seq, EV lumen, astrocyte-derived extracellular vesicles

## Abstract

Astrocytes are central nervous system (CNS)-restricted glial cells involved in synaptic function and CNS blood flow regulation. Astrocyte extracellular vesicles (EVs) participate in neuronal regulation. EVs carry RNAs, either surface-bound or luminal, which can be transferred to recipient cells. We characterized the secreted EVs and RNA cargo of human astrocytes derived from an adult brain. EVs were isolated by serial centrifugation and characterized with nanoparticle tracking analysis (NTA), Exoview, and immuno-transmission electron microscopy (TEM). RNA from cells, EVs, and proteinase K/RNase-treated EVs was analyzed by miRNA-seq. Human adult astrocyte EVs ranged in sizes from 50 to 200 nm, with CD81 as the main tetraspanin marker and larger EVs positive for integrin β1. Comparison of the RNA between the cells and EVs identified RNA preferentially secreted in the EVs. In the case of miRNAs, enrichment analysis of their mRNA targets indicates that they are good candidates for mediating EV effects on recipient cells. The most abundant cellular miRNAs were also abundant in EVs, and the majority of their mRNA targets were found to be downregulated in mRNA-seq data, but the enrichment analysis lacked neuronal specificity. Proteinase K/RNase treatment of EV-enriched preparations identified RNAs secreted independently of EVs. Comparing the distribution of cellular and secreted RNA identifies the RNAs involved in intercellular communication via EVs.

## 1. Introduction

Astrocytes are central nervous system (CNS)-restricted glial cells that tile the entire CNS [1,2], and as an integral part of the blood–brain barrier (BBB), they regulate the blood flow through the brain [3]. Their best known functions are to regulate synaptic formation and neuronal activity. In an adult brain, they show different morphologies and location-specific transcriptomes [4,5,6]. The most commonly used markers for astrocyte identification are glial fibrillary acidic protein (GFAP), S-100 calcium-binding protein (S100B), aldehyde dehydrogenase 1 family member L1 (ALDH1L1), glutamate transporter 1 (GLT1), aquaporin 4 (AQP 4), and Sox9 [7]; although, these proteins are also expressed by other cell types and do not represent all subpopulations of astrocytes [8].

Astrocyte communication with other cells in the brain is, in part, mediated by extracellular vesicles (EVs). Astrocyte EVs have, for instance, been shown to promote neuronal survival or death [9], dendritic growth, and synapse formation [10]. EVs carry protein, nucleic acids, and metabolites. RNA associated with EVs has been reported to be transferred to recipient cells and to mediate their responses [11,12,13]. EVs released in response to stimulation of astrocytes by IL-1β or TNF-α carry miRNAs which influence neuronal survival and differentiation in a rodent model [14]. Several secreted miRNAs have been shown to have functions such as the modulation of dendritic size and dendritic density [15], regulation of long-term potentiation and neuronal differentiation [16,17,18,19,20], and myelination [21].

Integrin β1, neuroprotective proteins such as apolipoprotein E (ApoE) and heat shock proteins have been identified in astrocyte EVs proteomics studies [22,23]. Specific EV cargoes have also been associated with pathological conditions like Alzheimer’s disease [23,24], stroke, ALS [25], and viral infections [26,27] and have been shown to mediate some of the effects of the diseases [28]. Astrocyte EVs were also shown to cross the BBB to recruit leukocytes to the brain in a model of brain injury [29].

Studies characterizing astrocytes from the human brain and their extracellular vesicles are limited and whether the cells used are fetal or not is not always specified. Most EV studies use fetal astrocytes which proliferate in the developing brain and under normal cell culture conditions [9,10,14,30,31,32,33,34,35]. On the other hand, most adult astrocytes only proliferate in the case of reactive astrogliosis during infection [36,37] and injury [38]. Even rodent studies mostly use cells from fetal or neo-natal animals.

In this study, we investigated the RNA distribution in astrocyte and astrocyte EVs from a human adult brain. The use of EV-depleted fetal bovine serum (FBS) is often preferred to the use of serum-free collection medium with primary cells [9,22,39] but evidence suggests that complete EV removal from undiluted FBS is not feasible [40,41] and astrocytes derived from adult donors only require FBS for proliferation but not for survival. We therefore chose an FBS-free medium for the EV collection and characterized the effects of FBS depletion on the astrocytes’ transcriptome using mRNA-seq. We found that genes downregulated by FBS removal were associated with cell proliferation and cell adhesion, while enrichment for astrocytic function was observed for upregulated genes.

The EV surface is charged and attracts a large number of particles (proteins, nucleic-acid-binding proteins, RNA, and DNA) to form a corona [42,43]; as a result, extracellular RNA can be found not only on the surface and as intraluminal cargo of EVs, but also as bound to proteins [44,45] and lipoproteins [46]. To identify extracellular RNA encapsulated in EVs, two methods have been developed: one uses proteinase K to remove the protein aggregates and the EV corona, followed by RNase digestion to remove unprotected RNA, with EV luminal RNAs being protected by the lipid membrane [47,48]. Alternatively, detergent can be used to destroy EVs prior to RNAase treatment to identify RNA bound to protein [49].

Comparing cellular, EV-enriched, and EV-luminal RNA, we could identify a set of RNAs which were preferentially secreted as EV luminal cargo and may mediate the reported effects of astrocyte EVs on other neural cells.

## 2. Materials and Methods

### 2.1. Poly-L-lysine Coating of Culture Plates

Fifteen µg/mL poly-L-Lysine hydrobromide (Sigma Aldrich, St. Louis, MO, USA; P9155-5MG) in sterile water was added to the culture plates. The plates were incubated at 37 °C overnight and washed three times with sterile water before use for the cell culture.

### 2.2. Astrocytes Culture

Astrocytes from a 27-year-old Caucasian male were purchased from Innoprot, Derio, Spain (Cat.no. P10251), spun, and grown on poly-L-Lysine hydrobromide coated flasks. The cells were cultivated in astrocyte medium (Innoprot, Derio, Spain; P60101) with 2% fetal bovine serum, 1% astrocyte growth supplement, and pencillin/streptomycin solution. The cells were washed and maintained in serum-free astrocyte medium after reaching 80% confluency. Conditioned medium was collected twice a day, for several days, for EV isolation.

### 2.3. RNA Isolation from the Astrocyte (Cells)

The astrocytes were seeded in a six-well plate and switched to serum-free medium/full medium 24 h before RNA isolation. The medium was removed, and the cells were washed with 1× sterile PBS. RNA was isolated with an RNeasy mini kit (Qiagen, Hilden, Germany; cat.no:74104), using the protocol provided by the manufacturer, and a QIAshredder mini spin column (Qiagen, Hilden, Germany; cat.no:79656).

### 2.4. Library Preparation for Cell RNA Sequencing (Novogene, UK)

Fifty ng of RNA per sample was used as the input material for the RNA sample preparations using an NEBNext^®^ Ultra™ RNA Library prep kit for Illumina^®^ (New England Biolabs, Hitchin, UK). Paired-end libraries were sequenced on Novaseq 6000 150 bp.

### 2.5. Bioinformatics Analysis mRNA-seq

After removal of low-quality and adaptor bases, clean reads were aligned to human genome hg38 using STAR v2.6.1d (mismatch = 2), counted by FeatureCount v1.5.0-p3 (default setting) and differential analysis with DESeq2 v1.20.0 (padj ≤ 0.05). Enrichment analysis (GO, KEGG, DO, and Reactome) was performed with ClusterProfiler v3.8.1 (padj < 0.05).

### 2.6. EV Preparation

Astrocytes were cultured in complete medium and switched to serum-free medium 16 h before collection of the conditioned medium for EV isolation. The most commonly used EV-depleted FBS still had considerable amounts of EVs [40,41]; for these reasons, we opted to collect EVs in serum-free medium. The conditioned medium was passed through a 40 µm strainer and centrifuged at 300 g for 15 min at 4 °C to pellet the ells. The supernatant was centrifuged twice at 2000 g for 15 min at 4 °C, and filtered through a 0.8 µm MF-Millipore MCE membrane filter (Merck Life Science, Espoo, Finland. cat.no SLAA033SS), transferred to new tubes, and centrifuged in an AH-629 swing bucket rotor (Thermo Scientific, Vantaa, Finland) for 2 h at 100,000 g (23,600 rpm, raverage:11.655 cm and K factor at max. speed: 242). All of the pellets were resuspended in 50–200 μL of RNase-free 0.22 µm filtered 1× PBS. The purified EVs were quantified by nanoparticle tracking analysis (NTA) and characterized by ExoView. EVs were frozen slowly and stored at −70 °C. We submitted all relevant data from our experiments to the EV-TRACK knowledgebase (EV-TRACK ID: EV220407) [50].

### 2.7. Negative Staining for TEM

The EV sample was mixed gently and 5 µL of the sample was added to a Formvar-coated copper grid and incubated for 20 min. The grid was washed with 100 µL 1× PBS twice for 1 min. The sample was fixed by incubating the grid on a drop of 1% glutaraldehyde in PBS for 5 min. The grid was washed eight times for 1 min with a drop of water. The grid was stained with neutral 50 µL 2% uranyl acetate (UA) for 5 min and then incubated with 50 µL of 2% methylcellulose- 0.4% UA solution for 10 min on ice. The grids were air-dried for 10 min and stored in a grid storage box for imaging. Imaging was carried out using Tecnai G2 Spirit 120 kV TEM with Veleta and Quemesa CCD cameras.

### 2.8. Immuno-Negative Staining for TEM

The EV sample was gently mixed and a 3 µL droplet was incubated on a Formvar-coated copper grid for 20 min. The grid was washed with a 100 µL drop of 1× PBS twice for 1 min. The grid was washed once with 1% BSA in 1× PBS for 10 min. The grid was incubated with anti-integrin β1 antibody (Invitrogen, Waltham, MA, USA; cat.no: MA2910) 1:100 dilution for 20 min. The grid was washed three times with 1% BSA in 1× PBS, incubated with a 5 µL drop of anti-mouse IgG secondary antibody (Jackson Immunoresearch, Ely, UK) 1:2000 dilution for 20 min, washed three times for 1 min with 1% BSA in 1× PBS, incubated with a 5 µL drop of gold-conjugated protein A (10 nm) 1:200 dilution for 20 min, and washed three times for 1 min with a drop of 1× PBS. The sample was fixed by incubating the grid in a drop of 1% glutaraldehyde in PBS for 5 min and it was washed eight times for 1 min with a drop of water. The grid was stained with neutral 50 µL 2% uranyl acetate (UA) for 5 min and incubated with 50 µL of 2% methylcellulose and 0.4% UA solution for 10 min on ice. The grids were air-dried before imaging. Imaging was carried out using Tecnai G2 Spirit 120 kV TEM with Veleta and Quemesa CCD cameras.

The sizes of the integrin-β1-positive EVs were measured with ImageJ. A graph for the size distribution and Tukey;s post hoc test for paired comparison of means was generated with OriginPro 2022 (academic).

### 2.9. Nanoparticle Tracking Analysis

Nanoparticle tracking analysis (NTA) was carried out by using a NanoSight NS300 (NanoSight Ltd., Amesbury, UK) equipped with a 405 nm laser. All of the starting materials were diluted in double-distilled H_2_O. A minimum of three 60 s videos recorded for each sample with a camera level of 14 and a detection threshold of 3, as well as the temperature, were monitored throughout the measurements. Data analysis was carried out with NTA software version 3.1. (Build 3.1.46) to define the concentration and size of the measured particles with the corresponding standard error. For analysis, auto-settings were used for the blur, minimum track length, and minimum expected particle size.

### 2.10. Exoview R100 (Nanoview Biosciences, USA)

For single EV analysis using Exoview (Nanoview Biosiences, Brighton, MI, USA), EVs were quantified using NTA, and then diluted with solution A (Nanoview Biosciences, Brighton, MI, USA) to the concentration of 1 × 10^8^ particles/mL. Diluted EV samples were incubated with human tetraspanin or human tetraspanin cargo chips (imprinted with capture antibodies for CD 9, CD 81, CD 63, and the isotype control) overnight, and then the chips were washed with solution A three times for 3 min with shaking at 300 rpm. Fluorescent antibody solutions were made using the manufacturer-supplied blocking solution or a blocking solution for cargo, by adding 0.5 µL of fluorescently labelled anti-CD 9 (CF^®^ 488A), anti-CD 81 (CF^®^ 555), and anti-CD63 (CF^®^ 647). The chips were incubated with the fluorescently labelled antibodies for 1 hr and imaged using Exoview R100 Exoscan 2.5.5 software. The results were visualized and analyzed using Exoviewer 2.5.0 with fluorescent intensity thresholds of 300 for channels 555 and 647 and a threshold of 500 for channel 488.

For cargo analysis, the EVs bound to the chips were lysed with solution C and solution D (provided by the manufacturer) and incubated with fluorescently labelled antibodies: anti-CD9 (CF**^®^** 488A), anti-CD63 (CF**^®^** 647), and anti-syntenin 1 (CF**^®^** 555).

### 2.11. Protein Quantification

EV and cell lysate proteins were quantified using a Qubit™ broad range protein assay kit (Invitrogen, Waltham, MA, USA; Q33211)/PierceTM BCA assay (Thermo Scientific, Vantaa, Finland; 23225) following the manufacturer’s supplied protocol.

### 2.12. Proteinase K and RNase A/T1 Treatment of EVs

EVs were thawed slowly; 100 µL of EVs was treated with 3.3 µL of proteinase K (New England biolab, P8107S) at 37 °C for 30 min. The proteinase K was inhibited using 100 ng of Pefablac^®^ (Roche, Basel, Switzerland cat.no. 11429868001) mixed gently and incubated on ice for 30 min. Then, 0.5 µL of RNase A and 0.5 µL of RNase T1 were added. The mixture was incubated at 37 °C for 10 min. An aliquot of 10 µL was taken for TEM analysis. The rest was used for RNA isolation. This is a modified method adapted from the thesis of Dr. Ter-Ovanesyan [51].

### 2.13. RNA Extraction for the miRNA-seq

EV RNA extraction was carried out by using RNeasy**^®^** UCP minelute**^®^** columns (Qiagen, Hilden, Germany; cat.no: 74204). For the Qiazol**^®^** lysis reagent (Qiagen, Hilden, Germany; cat.no. 79306), 700 µL was added to the EVs and proteinase K- and RNase-A/T1-treated EVs and mixed followed by the addition of 90 µL of chloroform, mixed 15 times by shaking the tube up and down. The mixture was incubated at RT for 2 min and centrifuged for 15 min in 12,000× *g* at 4 °C. The upper phase was carefully removed and added to 800 µL of absolute ethanol and mixed by inversion. The solution was added to minelute^®^ columns spun 15 s at 12,000 rpm; the flowthrough was discarded. The column was washed with manufacturer-supplied RWT (700 µL, once) and RPE buffer (500 µL, twice). To evaporate the residual ethanol in the columns, the lid of the column was cut, the collection tube was changed, and open columns were centrifuged at 14,000 rpm for 5 min. RNA was eluted with 14 µL of RNase-free water by centrifuging it at 14,000 rpm for 1 min after 1-min incubation at room temperature. For cell RNA isolation, the cells were detached with 0.5× trypsin and then resuspended in DMEM10% FBS and centrifuged for 1 min at max speed; the supernatant was removed and the dry pellet was frozen at −70 °C until extraction. The same extraction method was used for RNA isolation as for EVs, except the elution step in 20 µL H_2_O. The RNA profile was analyzed on bioanalyzer chips pico6000 (Agilent, Santa Clara, CA, USA).

### 2.14. miRNA-seq

RNA from the cells and EVs was sequenced at the Finnish Functional Genomic Center (University of Turku). Five ng of RNA was used with a QIAseq miRNA library kit (Qiagen, Hilden, Germany) according to the manufacturer’s recommendation and sequenced on NovaSeq6000 SP v1.5 with a read length of 1 × 75 bp (single end, two lanes). Raw reads were processed with a custom Snakemake [52] miRNA pipeline [53]. Adapter sequences were removed from raw fastq-files, followed by low-quality trimming using cutadapt [54]. High-quality reads were then aligned with bowtie [55] against the high confidence tRNA sequences hg38 database from GtRNAdb [56] and the complete Escherichia phage phiX174 genome, as downloaded from NCBI under accession number NC_001422 to reduce contamination and noise. De-noised reads were aligned with STAR, v 2.7.3a [57] against the mature and hairpin miRBase databases, version 22.1 [58], which were then filtered for hsa (human) prior to alignment. Here, multi-mapped reads were assigned randomly to the primary alignment and only those were counted in the miRNA quantification. The pipeline performs additional steps, which were not used in this analysis, however.

For the hierarchical clustering and principal component analyses, the count data were standardized with respect to the sample-wise alignment rate against the mature miRNA database, using counts-per-million (CPM). Furthermore, for two samples A and B, the distances in the hierarchical clustering were calculated as “1—Pearson correlation (A,B)”. Differential miRNA expressions were tested for with genewise negative binomial generalized linear models with quasi-likelihood tests, implemented in edgeR’s function glmQLFit [59]. Herein, pair-wise group comparisons were performed, with a model without interception; estimated dispersion and norm factors were calculated with the TMM method. The test results were corrected for multiple testing with the Benjamini–Hochberg method [60].

Group-wise expressions were visualized using classical boxplots. Here, the inner box represents the inner 50% of the data (the interquartile range, IQR), as the upper and lower borders of the box are defined as the upper respective lower quartile. Within the box, the median is still indicated, and the two whiskers indicate those two observations that deviate at most 1.5 times the IQR from the median. Any observation that is still further away from the median is considered as an outlier and is indicated with an individual dot.

### 2.15. miRNA Target Analysis

Tarbase v8, an experiment-based miRNA target database, was used for target prediction [61,62]. In cases when experimental data were not available, prediction-based databases, miRDB [63,64,65] and TargetScan human release 8.0, September 2021 [66,67], were used to find targets. The targets predicted by both the databases were taken for further analysis.

### 2.16. GO Enrichment Analysis

The GO enrichment analysis of the predicted miRNA targets from the miRNA-seq and the proteins identified using mass spectrometry were determined using the Cytoscape v3.9.1. plug-in BINGO [68]. The GO database from Gene Ontology [69,70] was downloaded and used as the reference database. The cut-off of the *p*-value < 0.05 and a node size above 10 were used for miRNA target GO analysis. The correction for multiple testing was carried out using Bonferroni family wise error rate correction [71].

## 3. Results

### 3.1. Cells Transcriptome Change in Response to the Removal of FBS

The cells we used are primary astrocytes from a young adult human male (Figure 1A). A large number of cells are needed for EV preparations and downstream analysis, but primary astrocytes derived from an adult brain grow very slowly and only for a limited number of passages. We therefore decided to make repeated EV collections from the same cells and to measure the effect of FBS depletion on the cell transcriptome using mRNAseq. For this, we used separate smaller cell samples instead of sacrificing the cells during EV collections. A 24 h time-point was selected as most switches in gene expression should be noticeable at this point.

mRNA-seq analysis showed 3354 genes were significantly upregulated in the FBS-free conditions compared to the control culture conditions (908 at least 2-fold and 722 protein-coding) while 3730 were downregulated (833 at least 2-fold and 763 protein-coding) and 21050 remained unchanged. Among the downregulated genes were GFAP and AQP 4, while GLT1, ALDH1L1, SOX9, and NMDA receptor subunit GRIN2A were upregulated (Appendix A). Enrichment analysis showed that KEGG pathways for downregulated genes (Figure 1B) included the cell cycle, DNA replication, and cellular senescence, while glutathione metabolism and Axon guidance were enriched in the significantly upregulated genes (Figure 1C).

GO annotations (Figure 2) mostly showed enrichment for biological processes linked to cell division (DNA replication and mitotic nuclear division) in the downregulated genes, while the most significant biological processes enriched in the upregulated genes included the regulation of lipid metabolic processes, locomotory behavior, negative regulation of neuron differentiation, axon development, neuron cell death (Figure 2A), and glial cell development. For the cellular component GO terms, chromosome, spindle, and focal adhesion were most enriched for the downregulated genes, and the synaptic membrane was most enriched for the upregulated genes (Figure 2B). Molecular functions enriched in the downregulated genes included cell adhesion, molecule binding, actin binding, and other processes linked to DNA replication activities and some cytoskeletal changes. The molecular function most enriched in the upregulated genes was RNA polymerase II proximal promoter sequence-specific DNA binding. Disease-associated enrichment (DO and DisGeNet) clearly indicated that the up-regulated genes were associated with neurological disorders and astrocytoma (Figure 2C,D). Although enrichment for the downregulated genes was systematically more significant than for the upregulated genes, removing FBS mainly downregulated the genes involved in cell division and upregulated those related to neural cell activity.

### 3.2. Astrocyte EV Characterization

Astrocyte EVs were collected twice a day (morning and evening) for several days in astrocyte medium without FBS, and three consecutive collections were pooled for EV isolation. According to the MISEV 2018 guidelines [48], our EV preparations should be considered as EV-enriched. The first pool (0–36 h) was used for miRNA-seq (four separate EV preparations). Mean particle sizes of 107.9 +/− 14.9 nm were measured by NTA from three different EV preparations (Figure 3a). The TEM showed both small and larger EVs, 50 to 200 nm in size (Figure 3b). The distribution of tetraspanins showed that the highest percentage of EVs were double positive CD63/CD81 (over 38%); then, 25.2% were positive for CD81 only, 11.64% were positive for CD63 only, 12.05% were triple CD9/CD63/CD81 positive, and 9.37% were CD81/CD9 double positive. The proportion of EVs positive for CD9 only was very low (Figure 3C); CD81 was the most represented tetraspanin, detectable on 68.75% of the EVs captured on the chip. A total of 40.64% of the captured EVs were positive for syntenin 1. Of the syntenin 1-positive EVs, 35.14% were syntenin 1/CD63-positive, 23.18% were syntenin 1/CD63/CD81-positive, 19.15% wer syntenin 1/CD63/CD9-positive, and 4.81% had syntenin 1 and all three tetraspanins (Figure 3d).

**Figure 2 genes-14-00853-f002:**
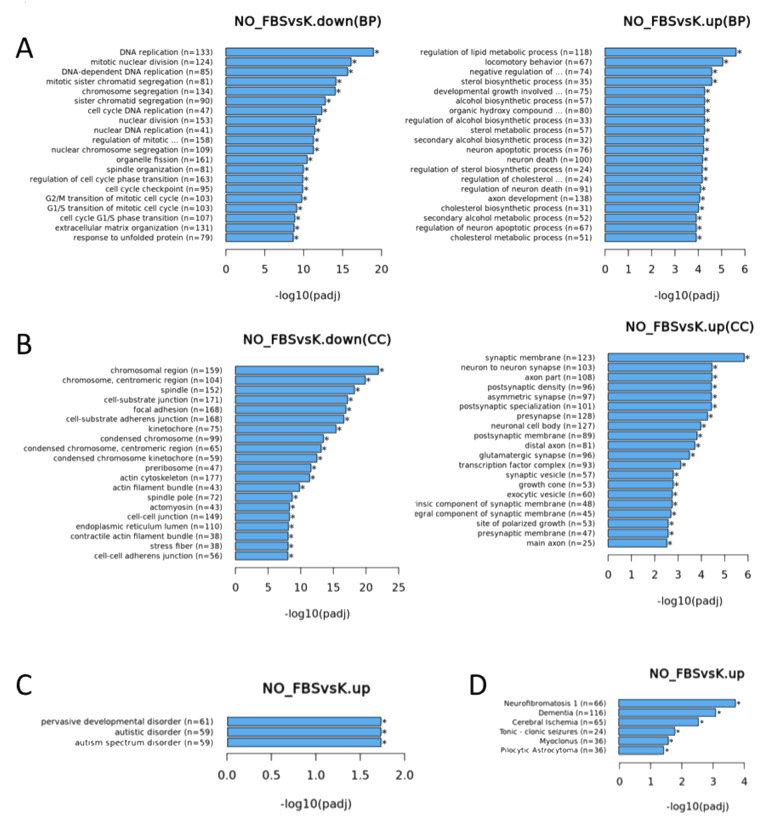
mRNA-seq (cells) GO annotations for differentially expressed genes (**A**) biological processes, (**B**) cellular component, down: downregulated genes, and up: upregulated genes). (**C**) DisGenNET of upregulated genes. (**D**) DO of upregulated genes. NO_FBS: astrocytes in culture for 24 h without 2% FBS, K: astrocyte for 24 h in full medium including 2% FBS. (NO_FBS—serum-free medium and K—medium with 2% FBS).

TEM images of integrin β1 immuno-stained EVs (Figure 4) were quantified with ImageJ; in total 106 EVs were counted and 71.6% were positive, with a mean size of 124.32 ± 81.33 nm, which was significantly bigger than the sizes of the integrin-β1-negative EVs of 65.55037 ± 36.36145 nm (*p*-value ≤ 0.001, using Tukey’s test for paired comparison of means).

### 3.3. RNA Profile

Extracellular RNA is transported by many types of particles including protein aggregates, lipoproteins, RNA-binding proteins, and EVs. To identify RNA transported in the EV lumen, we treated EV-enriched preparations with proteinase K followed by RNase A/T1.

Total cellular RNA and EV RNA were visualized by running 1 µL of purified RNA on a bioanalyzer chip. The cellular RNA profile showed small RNA peaks in addition to two peaks corresponding to rRNA. The RIN values were eight and above indicating good RNA quality (Figure 5a). The untreated EVs’ RNA showed a broad distribution, between 25 and 1000 bp with an occasional peak in the 1000 bp area (Figure 5b). After proteinase K and RNase A/T1 treatment, the bulk RNA distribution was restricted to the 25–200 bp area (Figure 5c) indicating the presence of a high proportion of unprotected RNA over a wide size range. TEM images showed intact EVs, mostly small (Figure 5d).

The miRNA-seq analysis started with quality trimming of the raw data. In this case, more short and poor-quality reads were removed from the EVs’ datasets than from the cells’ datasets (18% from cells, 35% from EV-enriched, and 47% for proteinase K/RNaseA/T1 treated EVs). Alignments against the reference genome hg38 and miRNA database resulted in different RNA biotype distributions between the cells and EVs, while the proteinase K- and RNase-A/T1-treated EVs and the untreated EVs had a more similar distribution. The proportion of unmapped reads was higher for EV RNA, while the number of unassigned_reads due to ambiguity was higher for cellular RNA. Although the sequencing method is designed to preferentially target miRNA and piRNA, the proportion of miRNA in the cells was just above 30% of the reads, while it was less than 10% in the EVs. However, the proportion of unassigned_NoFeature reads (unannotated) was close to 30% in the EVs but no more than 10% in the cells. LincRNA, protein-coding RNA, and tRNA were more represented in the EVs (Figure 5e). SnoRNA and processed-transcript biotypes were only detected in very limited amount in the EVs compared to the cells. We performed a more detailed analysis of the miRNAs.

### 3.4. miRNA Analysis

The clustering (Figure 6a) and PCA showed clear separation between the cells (group 1) and EV miRNA data (Figure 6b), with less clear separation between the EV RNA (group 2) and proteinase K- and RNase-A/T1-treated EVs (group 3).

A total of 406 miRNAs were identified, from which 207 were unchanged between the cells and the EVs (treated and untreated), 80 miRNA had different expression profiles between the cellular and untreated EVs, 83 miRNA had different representation between the cells and the proteinase K- and RNase-A/T1-treated EVs, and 83 were also significantly different between the EVs and the proteinase K- and RNase-A/T1-treated EVs (199 miRNAs were significantly different between at least two of the three groups). We identified 27 miRNAs preferentially secreted and protected from RNase A/T1 degradation (Table 1). The top 15 are presented in Figure 7.

The most abundant cell miRNAs could be separated into: (a) miRNAs that were significantly more abundant in cells and (b) miRNAs not significantly different between the cells and EVs (Figure 8, Table 2).

The top cellular miRNAs (hsa-miR-let-7a, hsa-miR-let-7b, hsa-miR-let-7f, hsa-miR-125a-5p, hsa-miR-21-5p, and hsa-miR-125b-5p) were also highly represented in the EVs. Preliminary qPCR data from a different donor’s astrocytes confirmed that hsa-miR-148a-3p, hsa-miR-24-3p, hsa-miR-26a-5p, hsa-miR-99a-5p, hsa-miR-21-5p, hsa-miR-9-5p, and has-let-7b-5p are more abundant in cells than derived EVs (Appendix A). Performing proteinaseK/RNaseA/T1 treatment in the presence of 1% triton X100 resulted in a reduction to undetectable levels of tested miRNA (ct values above 35 for hsa-let-7b-5p, hsa-miR-24-3p, and hsa-miR-99a-5p). MiRNAs hsa-miR-100-5p, hsa-miR-92a-3p, and hsa-miR-155-5p were equally represented in the EVs and cells, and hsa-miR-221-3p and hsa-miR-29a-3p (among the top 15 cellular miRNAs) showed no significant difference between the cells and EVs but were significantly decreased by proteinase K and RNase A/T1 treatment. At the other end of the spectrum, has-miR-34a-3p and hsa-miR-2277-5p were only detected in cells but in low amounts. Six miRNAs (hsa-miR-3663-3p, hsa-miR-1914-3p, hsa-miR548b-5p, hsa-miR-5699-5p, hsa-miR-6863, and hsa-miR-4259) could only be detected in untreated EVs and in very low quantity (normalized count value <10) but were absent from cells and the proteinase K- and RNase-A/T1-treated EVs.

### 3.5. miRNA Target Analysis

We found 9592 mRNA targets for the top 15 preferentially secreted and protected miRNAs (Figure 7) from Tarbase v8, targetscan, and mirDB (Appendix A). Two hundred and sixty-seven GO annotations for biological processes were found using Bonferroni family wise error rate correction with minimum enrichment of 10 folds and an adjusted *p*-value < 0.05. The main GO biological process clusters identified were the regulation of gene expression, phosphorus metabolism, system development, and nervous system development, implying regulation of neurons via the control of biosynthetic and metabolic processes, neuroblast proliferation, and neurogenesis (Figure 9a). Sixty-three GO annotations for the cellular component were found for preferentially secreted miRNA targets, of which, vesicle, synapse, and neuron projection were the most prominent (Figure 9b). Of the 51 GO annotations for molecular function identified for preferentially secreted miRNA targets, nucleotide binding and transferase and kinase catalytic activity were the most enriched (Appendix A).

We found 10,280 mRNA target genes for the 15 top cellular miRNAs (Figure 10a) representing 334 GO annotations for biological processes (10-fold enrichment), with the main GO clusters being protein metabolism and nucleic acid metabolism. One hundred and six GO annotations for the cellular component were enriched for the cellular miRNA targets, including the nucleus, membrane-bound organelle, catalytic complex, and cytoskeleton (Figure 10b). There were eighty-four GO terms for the molecular function, of which, there were nucleotide binding, ion binding, and enzyme regulation (Appendix A).

The most obvious difference between the preferentially secreted in the EV lumen and the cell miRNA target’s GO biological process was the highly significant large number of nodes for nervous system development (Figure 9a and Figure 10a). While the preferentially cellular miRNA target’s GO annotation “nervous system development”was less significant with only three nodes compared with the target of the preferentially EV luminal miRNAs’ targets (Appendix A). In the GO cellular component analysis, neuron projection and synapse were only found in the preferentially secreted miRNA targets.

The top 15 cellular miRNAs were better characterized with more reliable experimental data available in Tarbase v8 (Appendix A). In contrast, 6 of the top 15 preferentially secreted in the EV lumen miRNAs had no experimental data available in Tarbase v8. We had to use the targets found by both Targetscan and miRDB for the following: hsa-miR-203b-5p, hsa-miR-4497, and hsa-miR-7704. For the miRNAs, hsa-miR-3195, and hsa-miR-4492, there were no targets found using miRDB, so only targets from Targetscan were used for analysis. Only targets from miRDB could be found for hsa-miR-12136 (Appendix A). Targetscan and miRDB showed targets for miRNAs found using target prediction software, which was less reliable than those acquired with experimental evidence.

### 3.6. Exonic RNA

Exonic RNA includes protein-coding RNA and many species of non-coding RNA, including unprocessed and partly processed miRNA precursors, which can be identified both from cells and secreted vesicles. Other small non-coding RNAs include SNORD (60–150 nucleotides), snRNA (also in the 150 nt range), miscellaneous RNA, such as RNY1, and many other ribosome-associated RNAs which are in the 100 bp range. In our data, SNORDs were preferentially cellular with a small proportion, usually less than 10%, secreted as EV luminal cargo (Appendix A). The type of sequencing we used preferentially targeted smaller species such as mature miRNA, but the presence of longer RNA encapsulated in the EVs means that mRNA could be detected and identified. Two hundred and fifty-five mRNAs were only detected in EVs, and four hundred and thirty mRNAs were significantly enriched in proteinase K- and RNase-A/T1-treated EVs compared to the untreated EVs (Table 3). The most abundant are shown in Figure 11a. GO analysis in g:profiler (ordered) [72] was performed for the EV luminal mRNA fraction; all of the mRNAs with RPKM ≥ 10 in group 3 were included. The most enriched molecular function was ATP binding (Figure 11b).

### 3.7. Unannotated Transcripts

Novel transcripts with no existing annotations were also identified and compared between the three groups. We found 461 unannotated transcripts with significantly different expressions in at least two groups (*p* < 0.05). Four hundred and twenty-three were significantly different between the cells and EV fractions, three hundred and thiry-eight were significantly different between the cells and EV lumen, and one hundred and nine were significantly different between the EVs and EV lumen. A comparison of the unannotated RNA between the cells and EVs showed that the six most abundant RNAs in the EVs were very low in the cells (Figure 12A); the most abundant unannotated RNA in the cells also tended to be less represented in the EVs with some exceptions (Figure 12B). Unannotated RNA required further manual evaluation, after which we selected the most abundant which did not overlap with the tRNA, rRNA, or any other annotated gene in the UGC genome browser (Table 4).

## 4. Discussion

In line with previous data, removal of FBS from astrocytes in cell culture does not affect their survival. This finding is in accordance with adult astrocyte phenotypes because adult astrocytes do not proliferate; they support neuron survival and activity in the CNS. Comparing cellular, EV-enriched preparations, and EV luminal RNAs using miRNA-seq, we identified a selection of miRNAs preferentially secreted and protected from proteinase K and RNase A/T1 treatment, indicating that they are likely transported inside EVs. Even though this should be taken with some caution because the preferentially secreted miRNAs included several miRNAs without experimentally validated targets (Appendix A), GO annotations of their target mRNA point to preferentially secreted miRNAs protected inside EVs as being good candidates to drive the reported effects of astrocyte EVs in neuronal processes, such as cell survival and axon regeneration [9,73]; miR-203 has been reported to play a role in neuronal cell death and the regulation of neuronal activity [74], and miR-205 has been reported to modulate the expression of the leucine-rich repeat kinase 2 (LRRK2) gene which is involved in Parkinson’s disease [75]. MiR-21, involved, among other roles, in neuronal repair and transferred by EVs in the brain [7,13,76] is not significantly different between cells and the EV lumen and miR-23, which is equally distributed in cells, and EVs are involved in myelination and neuronal differentiation [15,16]. Most of the preferentially secreted miRNAs are reported to be involved in cancer metastasis and viral infections but their potential role in the brain has not been investigated [77,78,79,80,81,82,83,84].

The top 15 cellular miRNAs included only well characterized miRNAs and one third of their targets were found to be downregulated in our experimental conditions according to the mRNA-seq data (Figure 13). It should also be noted that even though we have classified them as preferentially cellular, they are highly represented in the EVs as well; in this category we have miR-125a-5p and miR-16-5p which have been reported to play an important role in neuronal survival and differentiation [9], miR-9-5p which is known to modulate dendritic size and density [10], and miR-26a which is involved in the regulation of long-term potentiation and neuronal differentiation. In the case of miR-9-5p, miR-221-3p, and miR-29a-3p, we observed a significant reduction with proteinase K and RNase treatment, indicating they are also secreted independently of EVs. Most of these reports are from rodent models or, like Chaudhuri et. al. 2018, they characterized the cargo of astrocyte EVs with neurotrophic signaling but did not include cellular miRNAs or proteinase K/RNAse treatment in their studies for comparison. Extracellular miRNAs not associated with EVs have been reported to bind the cell surface receptors TLR7 and TRPA1 to induce neuronal sensory responses [85,86].

In contrast to miRNA, unannotated RNAs, which were highly represented in our data (20–30% in EVs), could be more clearly differentiated as cellular or secreted. Further analysis and experimental studies are needed to evaluate whether they are specific to the brain or astrocytes. However, as they have been reported to be better than miRNA for classifying clinical samples using extracellular RNA data, their characterization would be of interest [87]. One pitfall with unannotated RNA is that the GRCh38 genome we used for the alignment dates from 2013. The complete sequencing of the human genome was only finalized in 2022 [88], and annotation and update is a continuing process.

As the sequences are short, misidentification is also a potential issue. We found that some of our unannotated RNA overlapped partially with some tRNAs, while others were encoded by the opposite strand on rRNA genes that were both nuclear and mitochondrial.

Of the other RNA types, tRNA is the most represented type in EVs, as we observed earlier [89], but some may be overrepresented in the RNaseA/T1 protected fraction, because their double-strand conformation makes them resistant to RNase digestion [90]. Although the NGS approach chosen targets miRNA and small RNA of comparable size, we did detect other non-coding RNAs. C/D box small nucleolar RNA (SNORD) was preferentially cellular with a very small fraction in EVs (Appendix A). The proportion of protein-coding mRNAs was lower in the cellular fraction which is most likely due to the sequencing strategy: miRNAs are abundant in cells, whereas fragmented mRNA is unlikely to be found. It is difficult to evaluate the presence and abundance of intact mRNA in EVs but spliced mRNA can be isolated and characterized using classical mRNA-seq for low input [91]. Thus, it is not surprising to find spliced mRNA, intact or fragmented, in the EV lumen, as the only significant enrichment observed was for molecular function GO term ATP binding and protein binding; we cannot say that it is representative of cell type, although it is possible that only some sub-populations of astrocyte EVs carry representative mRNA which could be detected using longer read sequencing.

There are technical limitations to this study. Due to the nature of the cells (primary astrocytes derived from an adult brain), they grew slowly and only for a limited number of passages (<5). To limit their handling, we did not count them or measure cell death; as an alternative, we measured the effect of FBS removal on the cells’ transcriptome at a 24-h time-point in the cells grown separately in 6-well plates. This did not show increased expression of apoptosis genes, but as it was performed with separate cells, it might not reflect the behavior of the cells from which we collected the EVs. We used differential centrifugation for EV enrichment, and the EVs were not further purified to avoid loss. As a result, our protein/particle ratio was 1.84 × 10^8^, which is about 100-fold higher than the values expected for pure EVs [92]. To eliminate the contaminating proteins and to prevent RNA sticking to the EV surface, we used proteinase K and RNase A/T1 treatment of EV-enriched fractions, and we based our analysis on miRNA which was significantly higher in the treated EVs than in the cells.

We observed a loss of RNA and a distinct change in the bioanalyzer profile with proteinase K and RNase A/T1 treatment. We used the method reported in the thesis of D. Ter-Ovenysan [51], which was designed for plasma EVs; the only change we made was to use a water-soluble inhibitor of proteinase K. Plasma EVs are known to have a large corona. It is debated whether the corona is specifically part of the EV or whether there is random sticking of circulating proteins, meaning the method may have been harsh for astrocyte EVs collected from cell culture. Thus, we cannot rule out that some EVs were destroyed by the treatment [93]. We observed a bigger reduction in the longer RNAs with some reduction in the miRNA counts in the treated EVs. However, the treatment has the advantage in that it enriches EVs’ RNA, excluding protein/RNA complexes and most contaminants. The only contaminant carrying RNA that could resist are lipids and partly digested lipoprotein [94,95].

One other thing to consider is the effect of the culture on the primary astrocytes from an adult brain, where proliferation is limited [96]. As there are just a few available sources of human adult astrocytes, we could only report the results for cells derived from one donor.

## 5. Conclusions

Characterization of EVs secreted by human cortical astrocytes, derived from an adult brain, using single-EV-based methods showed that CD81 is the dominant tetraspanin while integrin β1 is characteristic of larger EVs. A comparison of cellular, secreted, and protected small RNAs identified RNA preferentially secreted and protected in EVs. While SNORDs appear to be preferentially cellular, specific miRNAs and unannotated RNA are clearly targeted to EVs. The preferentially secreted miRNAs are good candidates to mediate the effect of astrocyte EVs on neurons. Further studies are needed to determine the role and organ specificity of the unannotated RNAs we identified.

## Figures and Tables

**Figure 1 genes-14-00853-f001:**
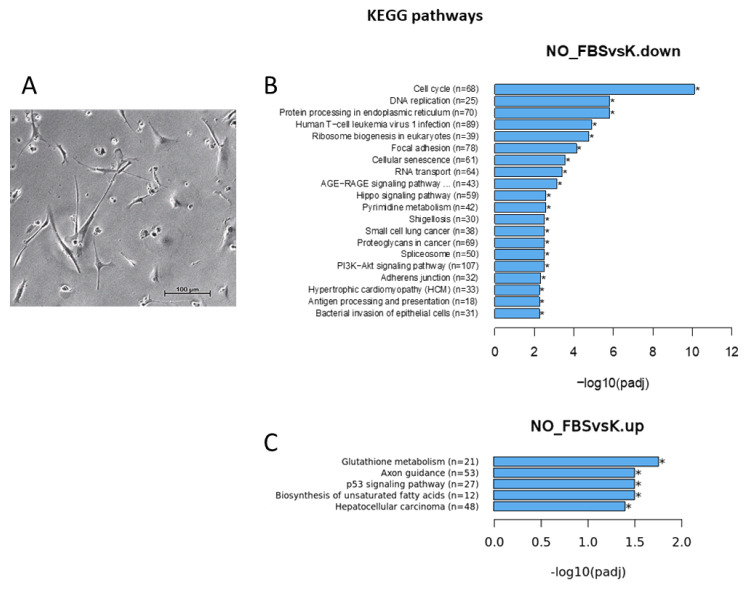
(**A**) Brightfield image of the astrocyte in culture using a 10× objective in Axiocam MRm attached to Axio Vert.A1 (Zeiss, Germany). KEGG pathways for (**B**) downregulated and (**C**) upregulated genes. (NO_FBS—serum-free medium and K—medium with 2% FBS).

**Figure 3 genes-14-00853-f003:**
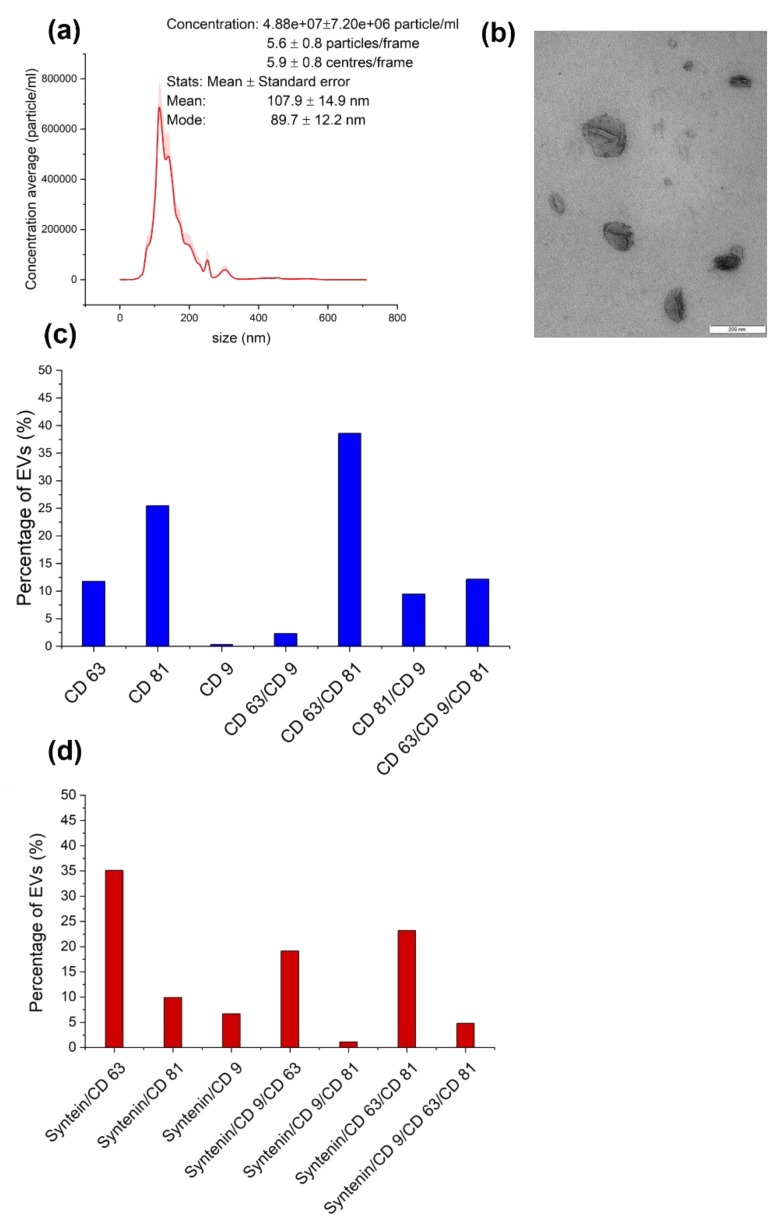
EV characterization (**a**) Nanoparticle tracking analysis (NTA) (Nanosight NS300) result from three different EV preparations at camera level 14 and detection level 3. (**b**) TEM negative staining of EV preparation. (**c**) Tetraspanin distribution measured by Exoview, expressed in the percentage of single, double, or triple positive tetraspanin EVs versus the total captured EVs. (**d**) Exoview, cargo analysis, and co-localization of syntenin 1 with a single tetraspanin and a combination of tetraspanins on single EVs, shown as a percentage of syntenin-1-positive EVs.

**Figure 4 genes-14-00853-f004:**
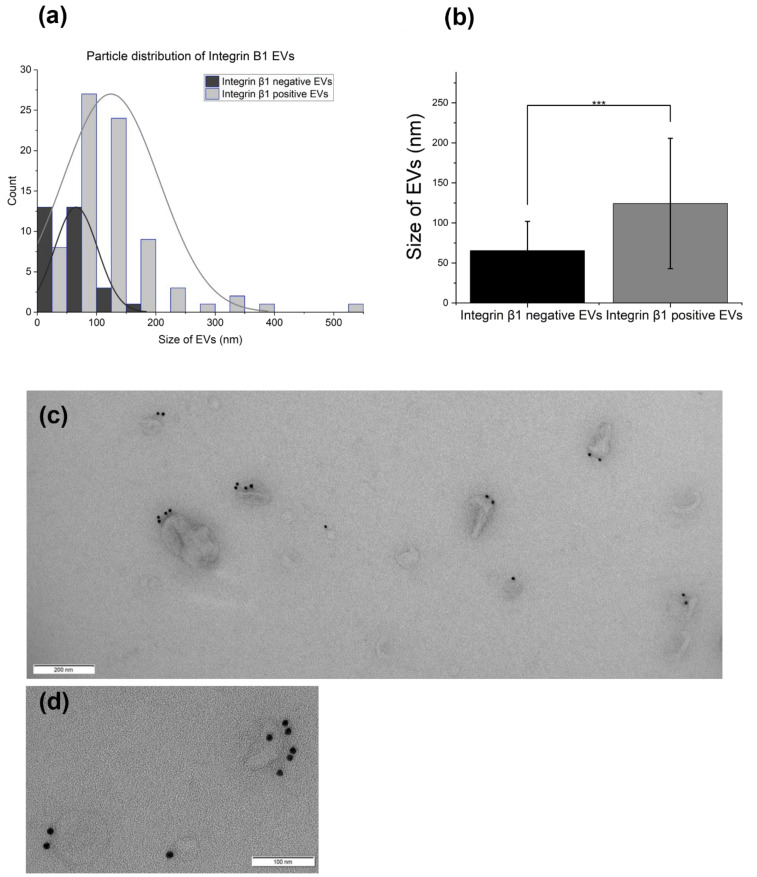
Integrin-β1-positive EVs: (**a**) size distribution of the EVs taken from TEM images of astrocyte EVs stained with an anti-integrin β1 antibody. Sizes measured with ImageJ. Graph generated by OriginPro 2022 (academic). (**b**) Graph indicating a significant difference in size between the integrin-β1-positive EVs vs. the negative EVs using Tukey’s test for paired comparison of the means. (**c**) TEM image. (**d**) Higher magnification TEM image.

**Figure 5 genes-14-00853-f005:**
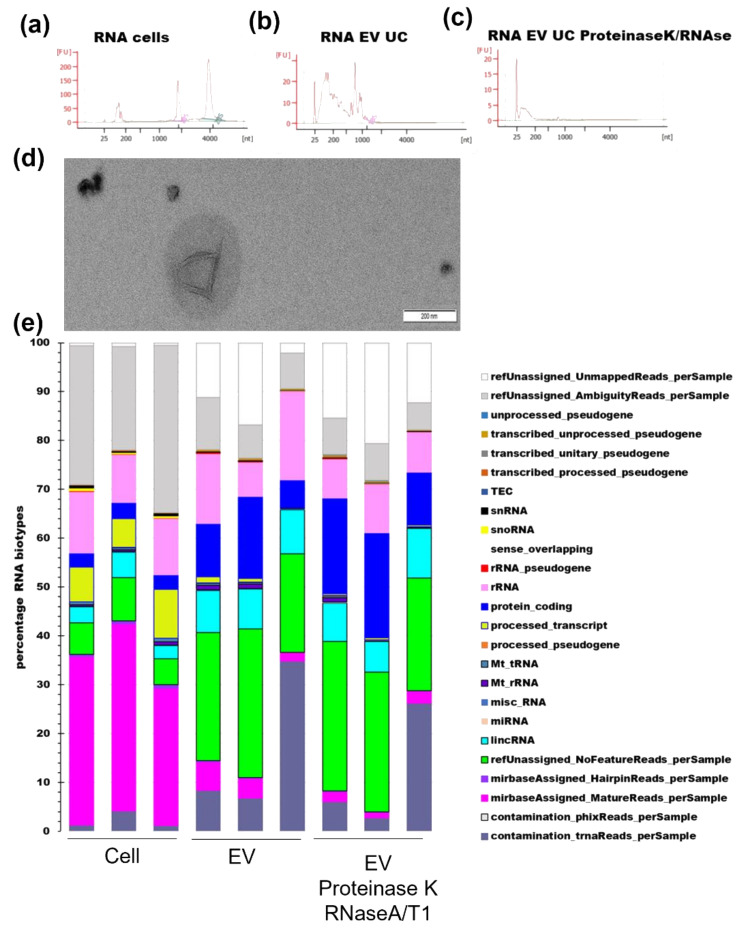
Bioanalyzer profiles of RNA extracted from (**a**) cells, (**b**) EVs, and (**c**) EVs treated with proteinase K and RNase A/T1. (**d**) TEM image of EVs treated with proteinase K and RNase A/T1. (**e**) miRNA-seq RNA biotype distribution per library.

**Figure 6 genes-14-00853-f006:**
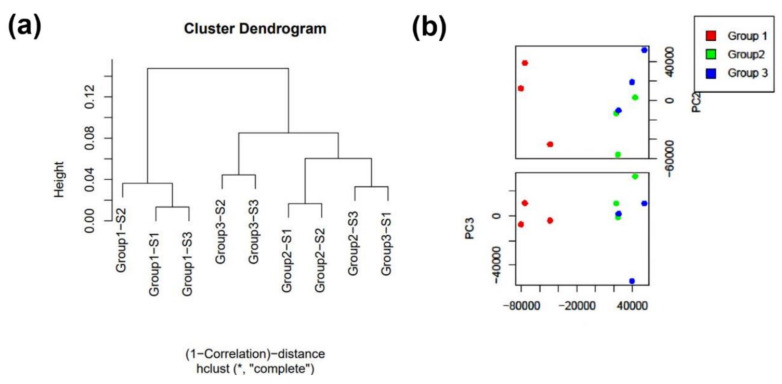
(**a**) miRNA-seq data analysis: cluster dendrogram for the miRNA data. (**b**) PCA (group 1—cells, group 2—EVs, and group 3—EV proteinase K and RNase A/T1).

**Figure 7 genes-14-00853-f007:**
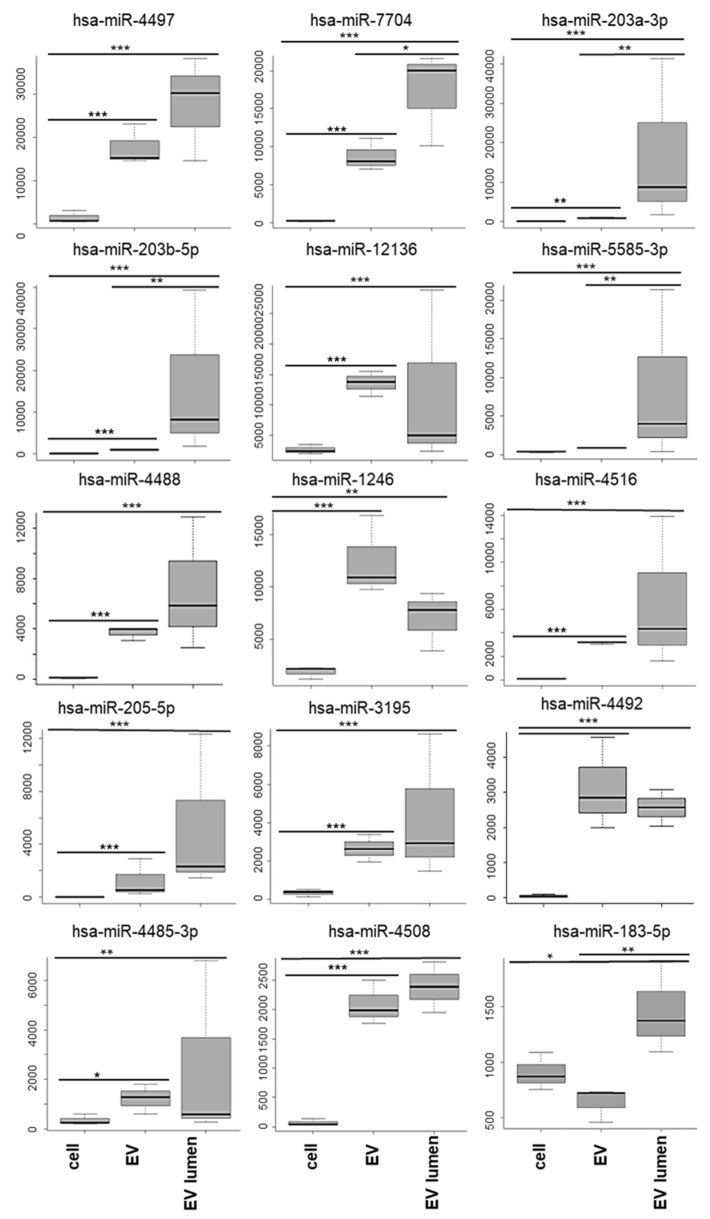
The 15 top miRNAs preferentially secreted in the EVs (mean TMM ± SD), Adj. *p* value: * <0.05, ** <0.001, and *** <0.0001. miRNA was used for the GO analysis.

**Figure 8 genes-14-00853-f008:**
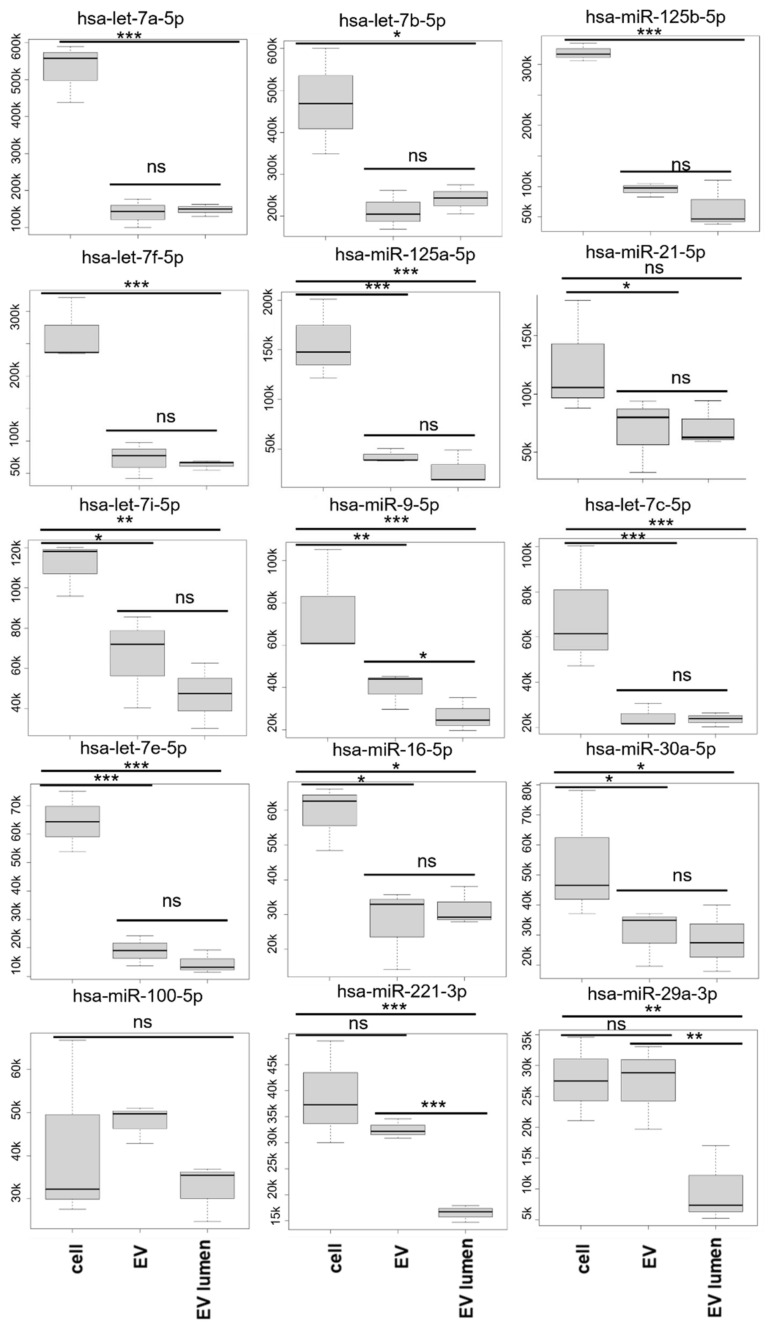
The 15 most abundant miRNAs in cells ordered from most to least abundant in cells. Mean TMM ± SD values. These miRNAs were used for the GO analysis.

**Figure 9 genes-14-00853-f009:**
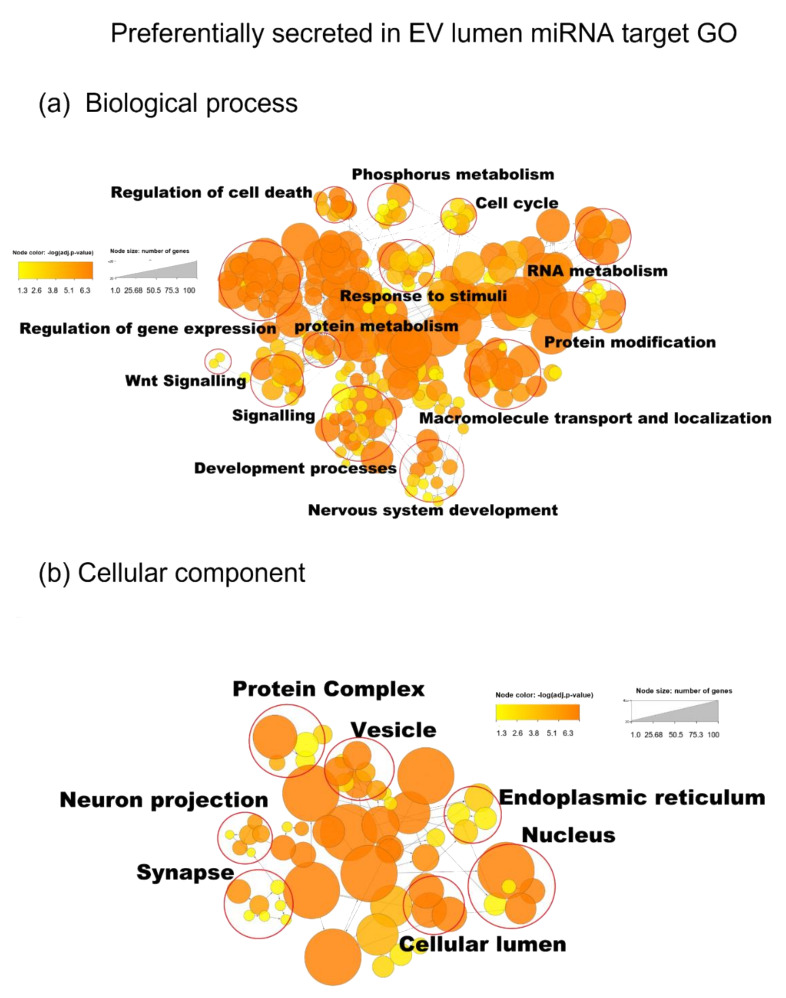
(**a**) GO annotations for biological process of the miRNA target of preferentially EV secreted miRNAs (from Figure 7) and (**b**) GO annotation for the cellular component of the miRNA target of the preferentially EV secreted miRNAs (from Figure 7). The analysis was performed using the BINGO app in Cytoscape_v3.9.1, with the adjusted *p*-value < 0.03 and the number of genes above 10.

**Figure 10 genes-14-00853-f010:**
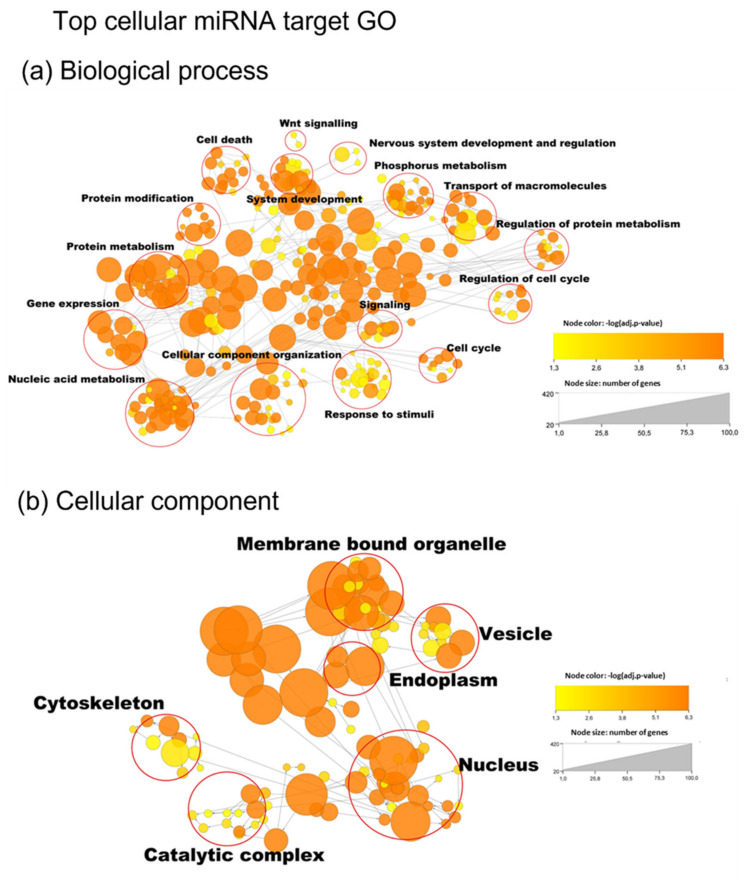
(**a**) GO annotation for biological processes of the cellular miRNA targets (from Figure 8). (**b**) GO annotation for cellular component of the cellular miRNA targets (from Figure 8). Analysis performed using the BINGO app in Cytoscape_v3.9.1, with the adjusted *p*-value < 0.03 and the number of genes above 10.

**Figure 11 genes-14-00853-f011:**
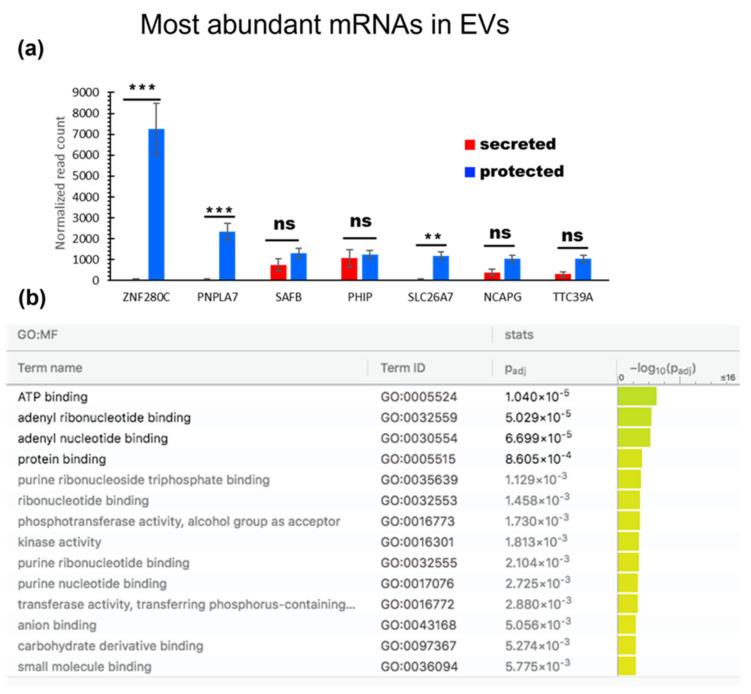
(**a**) Most abundant mRNA identified in the EVs. (**b**) GO annotations for the molecular function of the EV mRNA normalized value ≥10 from the g:profiler ordered query.

**Figure 12 genes-14-00853-f012:**
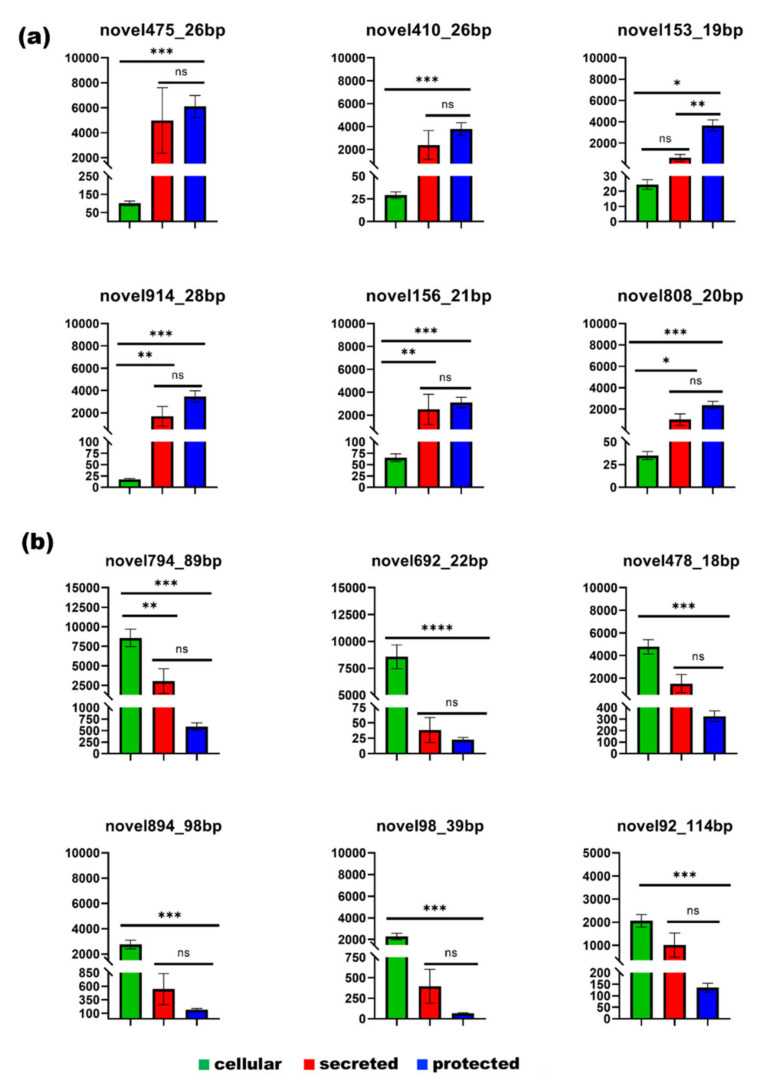
Most represented unannotated RNA: (**a**) the most abundant unannotated RNA in the EV lumen, and (**b**) the most abundant unannotated RNA in the cells (the chromosomal coordinates are in Table 4).

**Figure 13 genes-14-00853-f013:**
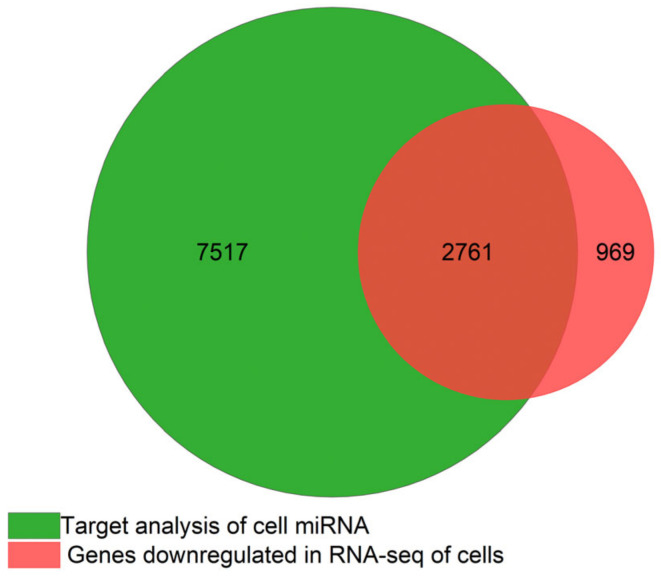
Venn diagram of the miRNA target from the top 15 cellular miRNA vs. RNA downregulated in mRNA-seq in astrocytes without FBS compared to the control grown in 2%FBS.

**Table 1 genes-14-00853-t001:** Preferentially secreted miRNAs and the *p*-values comparison for each group (1 = cellular, 2 = EV-enriched, and 3 = EV lumen), shown as the mean and SD of three separate biological replicates.

Preferentially Secreted miRNAs									
	*p*-Value			Cellular		EV		EV/Proteinase K/RNase A/T1	
	1 vs. 2	1 vs. 3	2 vs. 3	Mean	SD	Mean	SD	Mean	SD
**hsa-miR-4497**	3.44689 × 10^−6^	3.44689 × 10^−6^	0.318008849	1486.1	1445.9	17,683.5	4737.7	27,691.6	11,979.7
**hsa-miR-7704**	4.84641 × 10^−11^	4.84641 × 10^−11^	0.049571908	212.4	53.8	8732.5	2130.8	17,263.6	6224.7
**hsa-miR-203a-3p**	1.25252 × 10^−5^	1.25252 × 10^−5^	0.004314302	35.5	26.2	942.3	173.7	17,248.8	21,195.2
**hsa-miR-203b-5p**	4.3895 × 10^−6^	4.3895 × 10^−6^	0.005229131	10.4	8.1	879.7	33.4	16,387.7	20,067.6
**hsa-miR-12136**	0.003894963	0.003894963	0.799935098	2674.0	799.1	13,600.6	2087.0	12,101.3	14,566.1
**hsa-miR-5585-3p**	0.000119697	0.000119697	0.001698437	376.4	82.1	866.4	46.1	8608.8	11,229.4
**hsa-miR-4488**	7.06203 × 10^−9^	7.06203 × 10^−9^	0.111700189	126.9	44.7	3694.3	552.9	7084.3	5297.8
**hsa-miR-1246**	0.002486432	0.002486432	0.150556157	1836.4	581.3	12,459.7	3800.9	6997.8	2801.1
**hsa-miR-4516**	1.9021 × 10^−7^	1.9021 × 10^−7^	0.155468177	83.0	26.4	3167.3	102.0	6614.2	6458.7
**hsa-miR-205-5p**	2.21739 × 10^−5^	2.21739 × 10^−5^	0.121763527	7.2	7.9	1242.0	1463.1	5374.7	6033.5
**hsa-miR-3195**	3.41776 × 10^−5^	3.41776 × 10^−5^	0.299060818	342.1	195.9	2659.3	705.8	4340.8	3783.2
**hsa-miR-4492**	1.26325 × 10^−6^	1.26325 × 10^−6^	0.702245337	47.4	42.5	3135.9	1309.0	2561.7	518.3
**hsa-miR-4485-3p**	0.00215353	0.00215353	0.1993365	335.0	210.3	1217.2	601.2	2554.6	3692.2
**hsa-miR-4508**	1.14554 × 10^−5^	1.14554 × 10^−5^	0.809739857	71.8	60.6	2086.3	377.8	2384.9	429.1
**hsa-miR-183-5p**	0.131353291	0.044874832	0.001252323	904.3	168.4	636.8	156.3	1456.1	408.2
**hsa-miR-122-5p**	6.26532 × 10^−6^	3.03216 × 10^−7^	0.140718961	68.6	4.8	565.8	197.9	962.9	235.3
**hsa-miR-122b-3p**	7.42542 × 10^−5^	1.34355 × 10^−6^	0.067898761	62.1	7.5	397.4	123.8	815.3	186.1
**hsa-miR-219a-2-3p**	0.011896228	9.17169 × 10^−5^	0.041862396	79.2	37.6	243.6	15.1	593.8	345.5
**hsa-miR-3182**	0.00388248	0.027543506	0.280829595	58.1	49.5	1449.8	2038.1	541.5	843.3
**hsa-miR-320b**	0.377725867	0.046124912	0.232894024	14.8	18.4	207.5	18.7	350.6	157.3
**hsa-miR-320d**	0.377725867	0.046124912	0.232894024	142/8	18,4	207.5	18.7	350.6	157.3
**hsa-miR-320c**	0.059309372	0.014666686	0.487932701	90.7	44.5	217.7	50.6	299.5	56.7
**hsa-miR-8485**	0.021271633	0.006587555	0.543636599	33.5	23.8	181.2	153.4	275.9	271.7
**hsa-miR-196b-5p**	1.25523 × 10^−6^	2.15215 × 10^−8^	0.061851851	11.4	0.4	61.6	5.0	114.8	53.9
**hsa-miR-196a-5p**	0.007104663	0.000115384	0.082384637	19.5	8.6	50.1	30.6	97.3	46.8
**hsa-miR-219a-5p**	0.110077776	0.020295633	0.36931065	14.6	5.2	43.8	10.4	82.1	58.8
**hsa-miR-1269a**	0.110818989	0.006215975	0.13813471	5.2	1.4	19.8	12.3	69.9	54.4

**Table 2 genes-14-00853-t002:** The most abundant cellular miRNA identified (1 = cellular, 2 = EV-enriched, and 3 = EV lumen), shown as the mean and SD of three separate biological replicates. (The full table is shown as Appendix A).

miRNA	1 vs. 2	1 vs. 3	2 vs. 3	Mean	SD	Mean	SD	Mean	SD
**hsa-let-7a-5p**	5.83 × 10^−9^	1.22 × 10^−8^	7.30 × 10^−1^	529,093.3	79,765.6	139,857.4	38,543.4	147,506.1	16,764.8
**hsa-let-7b-5p**	1.21 × 10^−2^	3.17 × 10^−2^	6.63 × 10^−1^	472,859.8	126,401	212,171.8	46,930.1	241,446.2	34,655.4
**hsa-miR-125b-5p**	1.45 × 10^−6^	1.73 × 10^−8^	5.80 × 10^−2^	319,087.6	14,680.1	94,938.8	11,305.9	64,977.1	39,435.5
**hsa-let-7f-5p**	3.06 × 10^−7^	6.50 × 10^−8^	4.95 × 10^−1^	264,282.2	49,473.9	71,993.5	28,249.3	63,320.4	7203.6
**hsa-miR-125a-5p**	1.21 × 10^−4^	4.86 × 10^−6^	1.83 × 10^−1^	156,748.4	40,381.4	42,697.6	6905	29,103.6	17,366.3
**hsa-miR-21-5p**	4.29 × 10^−2^	6.01 × 10^−2^	8.70 × 10^−1^	124,441.8	48,925.7	68,901	31,859.7	72,126.8	19,283.1
**hsa-let-7i-5p**	2.63 × 10^−2^	6.53 × 10^−4^	1.33 × 10^−1^	111,401	13,359.8	66,078.8	23,112.2	46,871.5	16,231.5
**hsa-miR-9-5p**	2.05 × 10^−3^	9.43 × 10^−6^	4.29 × 10^−2^	75,573.1	25,632.3	39,619.9	8592.5	26,560.8	7953.5
**hsa-let-7c-5p**	1.41 × 10^−4^	8.59 × 10^−5^	8.43 × 10^−1^	69,692.5	27,534	24,635.8	5157.6	23,533.6	3101.8
**hsa-let-7e-5p**	1.42 × 10^−4^	1.34 × 10^−5^	3.34 × 10^−1^	64,413.8	10,642.7	19,035.4	5293.4	14,631.6	4079.7
**hsa-miR-16-5p**	2.10 × 10^−3^	9.51 × 10^−3^	5.34 × 10^−1^	59,081	9385.1	27,564.5	11,780.5	31,688.8	5533.6
**hsa-miR-30a-5p**	2.57 × 10^−2^	1.33 × 10^−2^	7.69 × 10^−1^	54,005.7	21,507.9	30,566.2	9554.5	28,467.3	11,094.4
**hsa-miR-100-5p**	6.40 × 10^−1^	3.27 × 10^−1^	1.54 × 10^−1^	42,245.5	21,385.5	47,892	4389.6	32,398.2	6672.9
**hsa-miR-221-3p**	3.12 × 10^−1^	4.54 × 10^−5^	6.31 × 10^−4^	38,967.4	9867.9	32,565.1	1863.6	16,469.9	1647
**hsa-miR-29a-3p**	9.44 × 10^−1^	1.52 × 10^−3^	1.80 × 10^−3^	27,723.2	6760.4	27,171.2	6834.6	9860.5	6266.4
**hsa-miR-127-3p**	4.78 × 10^−1^	7.10 × 10^−3^	3.51 × 10^−2^	25,280.3	6237.8	20,770.9	1214.4	11,263.9	5171.6
**hsa-miR-5701**	1.26 × 10^−3^	1.03 × 10^−5^	4.21 × 10^−2^	24,280.6	11,388.8	5983.5	4906.1	2657.1	960.8
**hsa-miR-99b-5p**	3.60 × 10^−3^	4.76 × 10^−4^	4.21 × 10^−1^	20,894.3	4077.9	10,705	2112.1	9030.7	3764.8
**hsa-miR-155-5p**	1.85 × 10^−1^	8.49 × 10^−2^	6.62 × 10^−1^	20,070.4	5918.6	11,339.5	5008.2	9422.3	4966.9
**hsa-miR-191-5p**	3.81 × 10^−2^	1.50 × 10^−3^	1.74 × 10^−1^	19,803.5	4049.2	11,828	771.8	8503.3	4741.3
**hsa-miR-31-5p**	1.14 × 10^−1^	6.65 × 10^−3^	1.89 × 10^−1^	19,435.5	6663.5	11,579.1	695.4	7556.1	5367.7
**hsa-miR-432-5p**	5.03 × 10^−2^	3.76 × 10^−2^	8.88 × 10^−1^	18,655.3	4168.3	10,851.8	2684.8	10,452.5	1212.2
**hsa-miR-181a-5p**	5.70 × 10^−2^	7.21 × 10^−5^	1.01 × 10^−2^	18,287.2	1595.9	12,258.9	642.7	7009.5	2056.9
**hsa-miR-26a-5p**	4.16 × 10^−3^	1.67 × 10^−3^	7.08 × 10^−1^	18,210.9	5627.8	9024	2513.7	8292.4	2714.5
**hsa-miR-92a-3p**	2.79 × 10^−1^	8.79 × 10^−1^	2.20 × 10^−1^	14,085.7	371.8	11,360	2628	14,518.7	8103.1
**hsa-miR-10a-5p**	5.25 × 10^−3^	6.48 × 10^−5^	8.66 × 10^−2^	14,036.5	4479.4	6903.2	1719.8	4567.8	1295.2
**hsa-miR-34a-5p**	6.21 × 10^−2^	8.50 × 10^−5^	8.73 × 10^−3^	14,017.5	4038.1	7588.1	616.6	3067.5	2179.7
**hsa-miR-30d-5p**	2.04 × 10^−3^	3.01 × 10^−4^	4.38 × 10^−1^	12,473.2	596.6	5033.3	2021.5	4092.6	2103.3
**hsa-miR-151a-3p**	1.32 × 10^−2^	1.66 × 10^−2^	9.22 × 10^−1^	11,823.7	2825.5	6597.2	2217.5	6739.2	1057.9
**hsa-miR-409-3p**	1.30 × 10^−1^	8.38 × 10^−3^	1.90 × 10^−1^	11,160.4	916	5990.6	871.8	3503.2	1113.7
**hsa-miR-222-3p**	2.55 × 10^−2^	2.61 × 10^−3^	3.33 × 10^−1^	10,942.3	1406.4	6227.5	2143.2	4934.5	1916.3
**hsa-miR-27b-3p**	3.56 × 10^−2^	1.14 × 10^−1^	5.61 × 10^−1^	10,782	970.9	5677.7	842.3	6720.4	3611.2
**hsa-miR-103b**	1.20 × 10^−1^	1.28 × 10^−4^	6.84 × 10^−3^	10,578	2875	7452.5	2397	3915.6	656.5
**hsa-miR-103a-3p**	1.13 × 10^−1^	2.21 × 10^−4^	1.22 × 10^−2^	10,515.6	2827.4	7346.5	2603.3	4064	591.1

**Table 3 genes-14-00853-t003:** Protein-coding RNA significantly different between at least two groups. Group 1: cells, group 2: EV, and group 3: EV proteinase K/RNase A/T1.

		*p*-Value			Cellular		EV		EV Proteinase K/RNase A/T1	
Ensemble ID	Gene Name	1 vs. 2	1 vs. 3	2 vs. 3	Mean	SD	Mean	SD	Mean	SD
**ENSG00000056277**	** *ZNF280C* **	3.05 × 10^−3^	3.94 × 10^−10^	6.50 × 10^−7^	2.63	0.10	49.32	19.04	7249.93	1221.95
**ENSG00000130653**	** *PNPLA7* **	7.47 × 10^−4^	3.45 × 10^−8^	2.02 × 10^−4^	1.33	0.05	65.26	25.19	2342.14	394.76
**ENSG00000160633**	** *SAFB* **	6.80 × 10^−5^	1.62 × 10^−6^	1.68 × 10^−1^	26.16	0.95	754.40	291.18	1314.36	221.53
**ENSG00000146247**	** *PHIP* **	2.59 × 10^−5^	4.96 × 10^−6^	4.90 × 10^−1^	13.87	0.51	1071.58	413.61	1246.43	210.08
**ENSG00000147606**	** *SLC26A7* **	5.66 × 10^−3^	1.86 × 10^−6^	1.40 × 10^−3^	2.03	0.07	51.81	20.00	1171.03	197.37
**ENSG00000109805**	** *NCAPG* **	1.70 × 10^−4^	2.28 × 10^−6^	8.47 × 10^−2^	9.54	0.35	387.89	149.72	1043.40	175.86
**ENSG00000085831**	** *TTC39A* **	3.15 × 10^−5^	4.13 × 10^−7^	6.54 × 10^−2^	2.72	0.10	311.02	120.05	1039.82	175.26
**ENSG00000142102**	** *PGGHG* **	3.45 × 10^−2^	1.11 × 10^−3^	1.40 × 10^−1^	59.88	2.18	400.42	154.55	866.71	146.08
**ENSG00000163608**	** *NEPRO* **	8.08 × 10^−3^	3.28 × 10^−4^	1.72 × 10^−1^	26.21	0.96	345.68	133.43	734.30	123.76
**ENSG00000147614**	** *ATP6V0D2* **	4.61 × 10^−4^	2.58 × 10^−4^	7.84 × 10^−1^	11.51	0.42	748.64	288.96	652.44	109.97
**ENSG00000112414**	** *ADGRG6* **	1.68 × 10^−5^	6.01 × 10^−6^	6.78 × 10^−1^	9.48	0.35	659.23	254.45	609.43	102.72
**ENSG00000172037**	** *LAMB2* **	1.56 × 10^−3^	1.71 × 10^−4^	3.23 × 10^−1^	11.42	0.42	358.42	138.34	581.29	97.97
**ENSG00000177888**	** *ZBTB41* **	3.31 × 10^−5^	1.30 × 10^−6^	1.39 × 10^−1^	1.00	0.04	213.25	82.31	563.31	94.94
**ENSG00000196781**	** *TLE1* **	7.06 × 10^−3^	1.62 × 10^−6^	5.13 × 10^−4^	0.00	0.00	7.00	2.70	561.30	94.61
**ENSG00000148734**	** *NPFFR1* **	1.08 × 10^−4^	1.90 × 10^−5^	4.39 × 10^−1^	4.98	0.18	399.44	154.17	524.36	88.38
**ENSG00000146054**	** *TRIM7* **	1.94 × 10^−4^	2.84 × 10^−5^	4.47 × 10^−1^	14.09	0.51	428.12	165.25	506.73	85.41
**ENSG00000213398**	** *LCAT* **	1.18 × 10^−4^	1.57 × 10^−5^	3.43 × 10^−1^	1.99	0.07	259.65	100.22	422.23	71.17
**ENSG00000137699**	** *TRIM29* **	7.20 × 10^−8^	5.26 × 10^−6^	1.06 × 10^−1^	9.28	0.34	1857.84	717.08	403.56	68.02
**ENSG00000177034**	** *MTX3* **	2.62 × 10^−1^	3.13 × 10^−4^	2.42 × 10^−3^	1.32	0.05	6.74	2.60	375.27	63.25
**ENSG00000272031**	** *ANKRD34A* **	2.37 × 10^−4^	2.21 × 10^−5^	2.54 × 10^−1^	1.32	0.05	176.66	68.19	363.44	61.26
**ENSG00000139641**	** *ESYT1* **	9.11 × 10^−4^	1.56 × 10^−4^	4.13 × 10^−1^	5.12	0.19	251.43	97.05	361.36	60.91
**ENSG00000124493**	** *GRM4* **	3.77 × 10^−3^	2.35 × 10^−5^	1.70 × 10^−2^	0.33	0.01	29.23	11.28	355.53	59.92
**ENSG00000156345**	** *CDK20* **	3.42 × 10^−1^	1.22 × 10^−5^	4.24 × 10^−5^	0.00	0.00	0.31	0.12	341.28	57.52
**ENSG00000049239**	** *H6PD* **	7.37 × 10^−2^	1.64 × 10^−3^	5.58 × 10^−2^	4.28	0.16	47.02	18.15	332.48	56.04
**ENSG00000103034**	** *NDRG4* **	1.30 × 10^−2^	1.04 × 10^−3^	2.33 × 10^−1^	8.88	0.32	152.32	58.79	327.29	55.16
**ENSG00000006715**	** *VPS41* **	4.75 × 10^−3^	8.03 × 10^−6^	3.87 × 10^−3^	0.00	0.00	10.03	3.87	325.90	54.93
**ENSG00000170190**	** *SLC16A5* **	7.64 × 10^−3^	2.18 × 10^−5^	5.66 × 10^−3^	0.00	0.00	9.65	3.72	325.37	54.84
**ENSG00000166135**	** *HIF1AN* **	2.23 × 10^−4^	1.84 × 10^−4^	9.32 × 10^−1^	6.88	0.25	440.75	170.12	322.07	54.28
**ENSG00000149577**	** *SIDT2* **	2.80 × 10^−6^	3.16 × 10^−6^	9.51 × 10^−1^	0.33	0.01	500.52	193.19	321.09	54.12
**ENSG00000163785**	** *RYK* **	6.33 × 10^−4^	8.99 × 10^−5^	3.52 × 10^−1^	2.61	0.10	193.18	74.56	319.40	53.83
**ENSG00000177380**	** *PPFIA3* **	1.95 × 10^−4^	2.17 × 10^−5^	3.13 × 10^−1^	1.99	0.07	187.81	72.49	318.46	53.68
**ENSG00000135870**	** *RC3H1* **	1.61 × 10^−1^	1.27 × 10^−3^	1.88 × 10^−2^	3.62	0.13	23.39	9.03	317.04	53.44
**ENSG00000183778**	** *B3GALT5* **	6.26 × 10^−2^	2.03 × 10^−3^	1.12 × 10^−1^	14.79	0.54	103.97	40.13	307.03	51.75
**ENSG00000114127**	** *XRN1* **	2.04 × 10^−3^	9.23 × 10^−5^	1.41 × 10^−1^	1.96	0.07	101.34	39.11	307.00	51.74
**ENSG00000120256**	** *LRP11* **	1.03 × 10^−4^	8.04 × 10^−6^	1.72 × 10^−1^	0.00	0.00	95.54	36.88	300.75	50.69
**ENSG00000164916**	** *FOXK1* **	4.81 × 10^−2^	1.10 × 10^−3^	6.02 × 10^−2^	3.33	0.12	46.46	17.93	298.61	50.33
**ENSG00000143552**	** *NUP210L* **	4.12 × 10^−2^	3.14 × 10^−5^	1.34 × 10^−3^	0.00	0.00	2.90	1.12	287.18	48.40
**ENSG00000171303**	** *KCNK3* **	2.19 × 10^−4^	5.36 × 10^−3^	1.90 × 10^−1^	33.05	1.20	1164.85	449.61	285.04	48.04
**ENSG00000147164**	** *SNX12* **	4.87 × 10^−5^	2.60 × 10^−5^	7.60 × 10^−1^	1.33	0.05	311.23	120.13	280.68	47.31
**ENSG00000180884**	** *ZNF792* **	3.04 × 10^−4^	1.70 × 10^−5^	1.65 × 10^−1^	0.67	0.02	102.19	39.44	278.11	46.87
**ENSG00000188315**	** *C3orf62* **	4.71 × 10^−3^	9.39 × 10^−5^	4.85 × 10^−2^	0.34	0.01	35.08	13.54	277.18	46.72
**ENSG00000159388**	** *BTG2* **	9.27 × 10^−3^	3.31 × 10^−4^	1.06 × 10^−1^	2.34	0.09	68.79	26.55	269.71	45.46
**ENSG00000121892**	** *PDS5A* **	2.52 × 10^−2^	1.05 × 10^−3^	1.33 × 10^−1^	6.47	0.24	86.81	33.51	267.16	45.03
**ENSG00000161850**	** *KRT82* **	4.69 × 10^−2^	1.74 × 10^−4^	7.98 × 10^−3^	0.33	0.01	8.74	3.37	261.01	43.99
**ENSG00000188112**	** *C6orf132* **	2.37 × 10^−2^	1.28 × 10^−4^	1.25 × 10^−2^	0.33	0.01	13.02	5.03	259.72	43.77
**ENSG00000204577**	** *LILRB3* **	7.32 × 10^−4^	1.65 × 10^−4^	4.88 × 10^−1^	3.68	0.13	201.86	77.91	258.14	43.51
**ENSG00000185651**	** *UBE2L3* **	6.44 × 10^−4^	2.16 × 10^−3^	5.66 × 10^−1^	10.38	0.38	640.17	247.09	255.17	43.01
**ENSG00000125843**	** *AP5S1* **	1.80 × 10^−4^	4.14 × 10^−5^	4.86 × 10^−1^	1.66	0.06	195.74	75.55	252.75	42.60
**ENSG00000057608**	** *GDI2* **	2.65 × 10^−2^	2.07 × 10^−3^	2.35 × 10^−1^	11.83	0.43	125.25	48.34	251.89	42.45
**ENSG00000206262**	** *FOXL2NB* **	3.40 × 10^−2^	1.01 × 10^−3^	8.67 × 10^−2^	3.94	0.14	56.18	21.68	251.27	42.35
**ENSG00000186185**	** *KIF18B* **	7.55 × 10^−4^	1.27 × 10^−5^	6.28 × 10^−2^	0.67	0.02	57.05	22.02	248.62	41.90
**ENSG00000171495**	** *MROH2B* **	4.27 × 10^−3^	1.18 × 10^−4^	9.65 × 10^−2^	2.02	0.07	68.24	26.34	246.60	41.56
**ENSG00000141837**	** *CACNA1A* **	2.20 × 10^−6^	3.45 × 10^−6^	8.37 × 10^−1^	0.99	0.04	428.96	165.57	245.22	41.33
**ENSG00000104967**	** *NOVA2* **	2.49 × 10^−1^	4.71 × 10^−3^	3.96 × 10^−2^	5.28	0.19	25.65	9.90	245.15	41.32
**ENSG00000077942**	** *FBLN1* **	5.86 × 10^−2^	2.24 × 10^−3^	1.06 × 10^−1^	5.88	0.21	60.71	23.43	242.10	40.80
**ENSG00000110324**	** *IL10RA* **	1.04 × 10^−1^	2.12 × 10^−4^	4.13 × 10^−3^	0.33	0.01	4.90	1.89	233.84	39.41
**ENSG00000136828**	** *RALGPS1* **	4.43 × 10^−1^	9.55 × 10^−3^	3.74 × 10^−2^	6.83	0.25	20.84	8.04	226.74	38.22
**ENSG00000122863**	** *CHST3* **	5.21 × 10^−2^	1.19 × 10^−3^	5.39 × 10^−2^	1.65	0.06	27.55	10.63	225.20	37.96
**ENSG00000162924**	** *REL* **	9.54 × 10^−2^	5.51 × 10^−4^	1.43 × 10^−2^	1.33	0.05	13.53	5.22	217.43	36.65
**ENSG00000102468**	** *HTR2A* **	1.72 × 10^−1^	3.28 × 10^−4^	3.55 × 10^−3^	0.33	0.01	3.45	1.33	213.83	36.04
**ENSG00000162735**	** *PEX19* **	7.04 × 10^−3^	9.13 × 10^−5^	2.40 × 10^−2^	0.00	0.00	12.61	4.87	210.82	35.53
**ENSG00000012048**	** *BRCA1* **	9.05 × 10^−7^	2.01 × 10^−6^	7.17 × 10^−1^	0.66	0.02	410.82	158.57	206.53	34.81
**ENSG00000107949**	** *BCCIP* **	6.48 × 10^−4^	4.98 × 10^−5^	1.91 × 10^−1^	0.33	0.01	73.54	28.38	204.79	34.52
**ENSG00000068976**	** *PYGM* **	1.15 × 10^−2^	3.92 × 10^−4^	8.71 × 10^−2^	0.98	0.04	39.03	15.06	202.92	34.20
**ENSG00000154822**	** *PLCL2* **	3.31 × 10^−3^	7.15 × 10^−5^	4.25 × 10^−2^	0.00	0.00	19.33	7.46	200.14	33.73
**ENSG00000102178**	** *UBL4A* **	5.80 × 10^−3^	9.51 × 10^−5^	3.10 × 10^−2^	0.00	0.00	14.36	5.54	198.06	33.38
**ENSG00000157322**	** *CLEC18A* **	1.73 × 10^−3^	2.50 × 10^−4^	3.27 × 10^−1^	1.33	0.05	103.60	39.99	196.40	33.10
**ENSG00000154655**	** *L3MBTL4* **	2.80 × 10^−2^	1.84 × 10^−4^	1.58 × 10^−2^	0.34	0.01	11.54	4.46	195.64	32.97
**ENSG00000167117**	** *ANKRD40CL* **	1.10 × 10^−3^	5.97 × 10^−5^	1.09 × 10^−1^	0.00	0.00	37.94	14.65	194.74	32.82
**ENSG00000175792**	** *RUVBL1* **	1.64 × 10^−3^	2.60 × 10^−4^	3.60 × 10^−1^	1.67	0.06	111.26	42.95	190.54	32.12
**ENSG00000226321**	** *CROCC2* **	6.60 × 10^−4^	4.99 × 10^−5^	1.89 × 10^−1^	0.33	0.01	68.25	26.34	190.11	32.04
**ENSG00000183813**	** *CCR4* **	2.78 × 10^−2^	3.52 × 10^−4^	2.70 × 10^−2^	0.33	0.01	13.36	5.16	189.62	31.96
**ENSG00000108576**	** *SLC6A4* **	2.84 × 10^−4^	1.03 × 10^−4^	6.09 × 10^−1^	1.00	0.04	166.01	64.08	187.12	31.54
**ENSG00000001461**	** *NIPAL3* **	1.01 × 10^−2^	5.94 × 10^−4^	1.46 × 10^−1^	1.33	0.05	51.41	19.84	185.44	31.26
**ENSG00000196664**	** *TLR7* **	1.74 × 10^−3^	2.57 × 10^−4^	3.24 × 10^−1^	1.01	0.04	94.64	36.53	184.81	31.15
**ENSG00000198626**	** *RYR2* **	1.86 × 10^−4^	8.75 × 10^−5^	7.06 × 10^−1^	0.99	0.04	184.95	71.39	181.56	30.60
**ENSG00000174945**	** *AMZ1* **	6.04 × 10^−3^	4.30 × 10^−4^	1.90 × 10^−1^	1.69	0.06	65.49	25.28	179.80	30.30
**ENSG00000186517**	** *ARHGAP30* **	5.88 × 10^−4^	8.50 × 10^−5^	2.62 × 10^−1^	0.00	0.00	66.31	25.59	178.92	30.16
**ENSG00000103197**	** *TSC2* **	7.61 × 10^−5^	2.86 × 10^−5^	6.32 × 10^−1^	0.66	0.02	163.90	63.26	174.99	29.49
**ENSG00000089101**	** *CFAP61* **	3.45 × 10^−3^	6.12 × 10^−4^	3.82 × 10^−1^	2.05	0.07	103.29	39.87	174.17	29.36
**ENSG00000136717**	** *BIN1* **	2.00 × 10^−2^	4.27 × 10^−4^	5.32 × 10^−2^	0.67	0.02	22.60	8.72	173.01	29.16
**ENSG00000112787**	** *FBRSL1* **	8.16 × 10^−2^	1.04 × 10^−3^	3.42 × 10^−2^	1.98	0.07	19.79	7.64	171.30	28.87
**ENSG00000115919**	** *KYNU* **	2.73 × 10^−4^	3.50 × 10^−4^	9.04 × 10^−1^	2.31	0.08	281.22	108.55	170.18	28.68
**ENSG00000136274**	** *NACAD* **	1.47 × 10^−4^	4.01 × 10^−5^	4.59 × 10^−1^	0.00	0.00	108.03	41.70	170.00	28.65
**ENSG00000171105**	** *INSR* **	7.69 × 10^−3^	8.57 × 10^−4^	2.76 × 10^−1^	2.63	0.10	80.60	31.11	169.47	28.56
**ENSG00000069188**	** *SDK2* **	1.30 × 10^−2^	1.23 × 10^−4^	1.75 × 10^−2^	0.00	0.00	7.77	3.00	168.30	28.37
**ENSG00000152556**	** *PFKM* **	7.70 × 10^−1^	9.28 × 10^−3^	1.51 × 10^−2^	4.25	0.15	7.34	2.83	164.95	27.80
**ENSG00000162367**	** *TAL1* **	2.43 × 10^−3^	1.24 × 10^−4^	1.26 × 10^−1^	0.33	0.01	41.87	16.16	164.15	27.67
**ENSG00000181378**	** *CFAP65* **	7.48 × 10^−5^	1.61 × 10^−4^	7.26 × 10^−1^	2.68	0.10	320.63	123.76	160.32	27.02
**ENSG00000155897**	** *ADCY8* **	7.68 × 10^−2^	1.48 × 10^−3^	4.15 × 10^−2^	0.99	0.04	14.74	5.69	160.22	27.00
**ENSG00000136869**	** *TLR4* **	2.58 × 10^−5^	5.35 × 10^−5^	7.24 × 10^−1^	0.99	0.04	325.94	125.80	159.68	26.91
**ENSG00000156869**	** *FRRS1* **	1.11 × 10^−3^	3.48 × 10^−5^	7.13 × 10^−2^	0.00	0.00	26.40	10.19	158.91	26.78
**ENSG00000047648**	** *ARHGAP6* **	3.56 × 10^−3^	7.42 × 10^−5^	4.34 × 10^−2^	0.00	0.00	16.60	6.41	158.42	26.70
**ENSG00000156959**	** *LHFPL4* **	1.51 × 10^−3^	1.41 × 10^−4^	2.11 × 10^−1^	0.33	0.01	56.41	21.77	157.45	26.54
**ENSG00000133056**	** *PIK3C2B* **	2.25 × 10^−3^	4.45 × 10^−4^	4.06 × 10^−1^	1.31	0.05	94.09	36.32	153.32	25.84
**ENSG00000068323**	** *TFE3* **	8.87 × 10^−3^	6.96 × 10^−4^	1.93 × 10^−1^	1.35	0.05	52.22	20.16	151.32	25.51
**ENSG00000169876**	** *MUC17* **	3.73 × 10^−4^	1.91 × 10^−5^	1.23 × 10^−1^	0.00	0.00	39.46	15.23	151.32	25.50
**ENSG00000131126**	** *TEX101* **	2.10 × 10^−7^	3.73 × 10^−5^	4.04 × 10^−2^	3.29	0.12	1148.45	443.28	150.42	25.35
**ENSG00000170390**	** *DCLK2* **	7.79 × 10^−3^	1.72 × 10^−4^	4.10 × 10^−2^	0.00	0.00	12.22	4.72	147.67	24.89
**ENSG00000168070**	** *MAJIN* **	5.51 × 10^−2^	2.05 × 10^−4^	6.09 × 10^−3^	0.00	0.00	2.63	1.01	146.68	24.72
**ENSG00000132874**	** *SLC14A2* **	3.35 × 10^−5^	1.73 × 10^−5^	7.17 × 10^−1^	0.00	0.00	146.04	56.37	145.22	24.48
**ENSG00000153551**	** *CMTM7* **	4.62 × 10^−3^	1.60 × 10^−4^	6.69 × 10^−2^	0.00	0.00	17.38	6.71	144.92	24.43
**ENSG00000181215**	** *C4orf50* **	6.94 × 10^−5^	3.87 × 10^−5^	7.65 × 10^−1^	0.34	0.01	154.72	59.72	140.95	23.76
**ENSG00000168961**	** *LGALS9* **	3.05 × 10^−3^	1.20 × 10^−4^	7.96 × 10^−2^	0.00	0.00	20.78	8.02	140.55	23.69
**ENSG00000168487**	** *BMP1* **	2.25 × 10^−2^	2.55 × 10^−4^	1.98 × 10^−2^	0.00	0.00	6.01	2.32	139.54	23.52
**ENSG00000169860**	** *P2RY1* **	1.55 × 10^−4^	3.37 × 10^−5^	3.98 × 10^−1^	0.00	0.00	80.33	31.00	139.53	23.52
**ENSG00000138079**	** *SLC3A1* **	1.51 × 10^−2^	8.95 × 10^−4^	1.42 × 10^−1^	1.00	0.04	35.48	13.69	137.33	23.15
**ENSG00000050767**	** *COL23A1* **	1.11 × 10^−1^	2.08 × 10^−4^	2.86 × 10^−3^	0.00	0.00	1.36	0.53	135.73	22.88
**ENSG00000183760**	** *ACP7* **	2.19 × 10^−2^	1.09 × 10^−3^	1.18 × 10^−1^	1.00	0.04	29.60	11.43	134.93	22.74
**ENSG00000188343**	** *FAM92A* **	2.70 × 10^−2^	2.27 × 10^−4^	1.48 × 10^−2^	0.00	0.00	4.83	1.86	134.14	22.61
**ENSG00000006788**	** *MYH13* **	2.99 × 10^−3^	8.27 × 10^−5^	5.95 × 10^−2^	0.00	0.00	17.61	6.80	133.07	22.43
**ENSG00000095777**	** *MYO3A* **	5.31 × 10^−3^	8.25 × 10^−4^	3.45 × 10^−1^	1.69	0.06	71.88	27.74	131.93	22.24
**ENSG00000114779**	** *ABHD14B* **	8.15 × 10^−4^	3.22 × 10^−4^	6.01 × 10^−1^	0.34	0.01	104.77	40.44	128.59	21.67
**ENSG00000103489**	** *XYLT1* **	5.14 × 10^−4^	8.12 × 10^−5^	2.96 × 10^−1^	0.00	0.00	53.71	20.73	125.75	21.20
**ENSG00000117020**	** *AKT3* **	1.75 × 10^−3^	1.60 × 10^−4^	1.72 × 10^−1^	0.00	0.00	32.16	12.41	125.07	21.08
**ENSG00000094631**	** *HDAC6* **	8.49 × 10^−4^	1.29 × 10^−4^	3.25 × 10^−1^	0.33	0.01	63.89	24.66	124.69	21.02
**ENSG00000215912**	** *TTC34* **	9.33 × 10^−4^	9.98 × 10^−5^	2.12 × 10^−1^	0.00	0.00	39.67	15.31	120.63	20.33
**ENSG00000155858**	** *LSM11* **	2.95 × 10^−3^	4.26 × 10^−4^	3.10 × 10^−1^	0.67	0.02	57.84	22.33	120.20	20.26
**ENSG00000176601**	** *MAP3K19* **	9.62 × 10^−4^	1.48 × 10^−4^	3.52 × 10^−1^	0.66	0.02	69.04	26.65	119.74	20.18
**ENSG00000177181**	** *RIMKLA* **	1.10 × 10^−2^	3.46 × 10^−4^	5.53 × 10^−2^	0.00	0.00	10.80	4.17	119.21	20.09
**ENSG00000131951**	** *LRRC9* **	9.00 × 10^−3^	1.25 × 10^−3^	2.72 × 10^−1^	0.66	0.02	45.51	17.57	118.81	20.02
**ENSG00000261649**	** *GOLGA6L7* **	5.73 × 10^−3^	8.41 × 10^−4^	2.96 × 10^−1^	0.66	0.02	51.27	19.79	118.80	20.02
**ENSG00000043591**	** *ADRB1* **	4.30 × 10^−3^	1.06 × 10^−3^	4.47 × 10^−1^	0.99	0.04	73.88	28.52	118.38	19.95
**ENSG00000181555**	** *SETD2* **	2.04 × 10^−2^	6.36 × 10^−4^	6.86 × 10^−2^	0.33	0.01	16.22	6.26	118.36	19.95
**ENSG00000157637**	** *SLC38A10* **	1.76 × 10^−3^	1.90 × 10^−4^	1.99 × 10^−1^	0.00	0.00	33.48	12.92	118.01	19.89
**ENSG00000140403**	** *DNAJA4* **	1.37 × 10^−2^	1.93 × 10^−3^	3.00 × 10^−1^	1.67	0.06	53.70	20.73	117.28	19.77
**ENSG00000073803**	** *MAP3K13* **	1.76 × 10^−2^	5.29 × 10^−4^	6.86 × 10^−2^	0.34	0.01	17.00	6.56	117.06	19.73
**ENSG00000196628**	** *TCF4* **	9.66 × 10^−6^	3.68 × 10^−4^	1.18 × 10^−1^	3.29	0.12	666.80	257.37	115.50	19.47
**ENSG00000011376**	** *LARS2* **	3.45 × 10^−4^	5.24 × 10^−5^	2.99 × 10^−1^	0.00	0.00	51.95	20.05	114.89	19.36
**ENSG00000172349**	** *IL16* **	7.06 × 10^−3^	2.80 × 10^−4^	9.83 × 10^−2^	0.33	0.01	23.81	9.19	114.77	19.34
**ENSG00000100401**	** *RANGAP1* **	3.11 × 10^−2^	1.96 × 10^−3^	1.39 × 10^−1^	0.99	0.04	26.26	10.14	113.94	19.20
**ENSG00000140287**	** *HDC* **	5.49 × 10^−4^	8.45 × 10^−5^	2.92 × 10^−1^	0.00	0.00	48.67	18.79	113.90	19.20
**ENSG00000105419**	** *MEIS3* **	4.49 × 10^−2^	1.92 × 10^−3^	1.03 × 10^−1^	1.31	0.05	22.99	8.87	112.87	19.02
**ENSG00000078900**	** *TP73* **	8.37 × 10^−4^	2.08 × 10^−4^	4.57 × 10^−1^	0.33	0.01	73.88	28.52	112.62	18.98
**ENSG00000171467**	** *ZNF318* **	1.20 × 10^−3^	2.58 × 10^−4^	4.08 × 10^−1^	0.33	0.01	65.92	25.44	112.07	18.89
**ENSG00000106031**	** *HOXA13* **	1.60 × 10^−1^	1.60 × 10^−3^	2.03 × 10^−2^	0.66	0.02	6.04	2.33	109.86	18.52
**ENSG00000135766**	** *EGLN1* **	8.43 × 10^−3^	2.12 × 10^−4^	4.88 × 10^−2^	0.00	0.00	10.67	4.12	108.86	18.35
**ENSG00000184735**	** *DDX53* **	2.62 × 10^−2^	1.32 × 10^−3^	1.00 × 10^−1^	0.33	0.01	16.59	6.40	107.49	18.12
**ENSG00000185271**	** *KLHL33* **	3.79 × 10^−3^	7.72 × 10^−4^	4.05 × 10^−1^	0.99	0.04	63.67	24.58	106.27	17.91
**ENSG00000129451**	** *KLK10* **	1.00 × 10^0^	5.35 × 10^−4^	4.63 × 10^−4^	0.00	0.00	0.00	0.00	105.86	17.84
**ENSG00000105289**	** *TJP3* **	3.46 × 10^−3^	2.40 × 10^−4^	1.31 × 10^−1^	0.00	0.00	21.56	8.32	105.60	17.80
**ENSG00000175806**	** *MSRA* **	5.22 × 10^−1^	7.25 × 10^−3^	2.02 × 10^−2^	1.31	0.05	3.93	1.52	103.76	17.49
**ENSG00000100916**	** *BRMS1L* **	3.44 × 10^−3^	4.39 × 10^−4^	2.66 × 10^−1^	0.33	0.01	41.76	16.12	103.26	17.40
**ENSG00000189350**	** *TOGARAM2* **	8.38 × 10^−1^	1.39 × 10^−2^	1.89 × 10^−2^	2.31	0.08	3.67	1.42	101.98	17.19
**ENSG00000143921**	** *ABCG8* **	1.27 × 10^−2^	7.10 × 10^−4^	1.22 × 10^−1^	0.33	0.01	21.46	8.28	101.96	17.19
**ENSG00000162174**	** *ASRGL1* **	2.58 × 10^−3^	3.31 × 10^−4^	2.75 × 10^−1^	0.33	0.01	43.39	16.75	101.68	17.14
**ENSG00000162409**	** *PRKAA2* **	4.24 × 10^−3^	9.02 × 10^−4^	4.00 × 10^−1^	0.66	0.02	57.18	22.07	101.60	17.12
**ENSG00000156886**	** *ITGAD* **	4.53 × 10^−5^	1.53 × 10^−4^	4.74 × 10^−1^	0.00	0.00	345.95	133.53	101.21	17.06
**ENSG00000105877**	** *DNAH11* **	1.17 × 10^−2^	1.32 × 10^−3^	2.34 × 10^−1^	0.66	0.02	34.95	13.49	100.05	16.86
**ENSG00000183808**	** *RBM12B* **	5.58 × 10^−5^	2.37 × 10^−5^	6.46 × 10^−1^	0.00	0.00	91.18	35.19	99.62	16.79
**ENSG00000121904**	** *CSMD2* **	1.86 × 10^−3^	1.62 × 10^−4^	1.71 × 10^−1^	0.00	0.00	26.45	10.21	99.47	16.77
**ENSG00000108950**	** *FAM20A* **	1.01 × 10^−5^	1.36 × 10^−5^	8.77 × 10^−1^	0.00	0.00	170.13	65.67	99.12	16.71
**ENSG00000183695**	** *MRGPRX2* **	1.13 × 10^−3^	1.40 × 10^−4^	2.40 × 10^−1^	0.00	0.00	35.12	13.56	99.11	16.70
**ENSG00000171109**	** *MFN1* **	1.10 × 10^−2^	7.66 × 10^−4^	1.51 × 10^−1^	0.34	0.01	23.85	9.21	96.43	16.25
**ENSG00000141696**	** *P3H4* **	9.92 × 10^−4^	1.01 × 10^−3^	9.92 × 10^−1^	1.70	0.06	142.54	55.02	95.92	16.17
**ENSG00000188452**	** *CERKL* **	2.54 × 10^−2^	5.21 × 10^−4^	5.12 × 10^−2^	0.33	0.01	11.96	4.61	92.00	15.51
**ENSG00000124120**	** *TTPAL* **	3.33 × 10^−4^	8.26 × 10^−5^	4.34 × 10^−1^	0.00	0.00	55.66	21.48	91.46	15.41
**ENSG00000145536**	** *ADAMTS16* **	9.76 × 10^−5^	5.18 × 10^−5^	7.26 × 10^−1^	0.00	0.00	92.20	35.59	91.36	15.40
**ENSG00000106404**	** *CLDN15* **	7.29 × 10^−3^	2.02 × 10^−4^	5.71 × 10^−2^	0.00	0.00	10.75	4.15	90.85	15.31
**ENSG00000166260**	** *COX11* **	2.87 × 10^−3^	3.52 × 10^−4^	2.18 × 10^−1^	0.00	0.00	26.32	10.16	90.38	15.23
**ENSG00000132326**	** *PER2* **	2.77 × 10^−3^	5.87 × 10^−4^	3.93 × 10^−1^	0.34	0.01	49.52	19.12	90.02	15.17
**ENSG00000110171**	** *TRIM3* **	1.33 × 10^−4^	7.89 × 10^−5^	7.67 × 10^−1^	0.00	0.00	94.90	36.63	89.56	15.09
**ENSG00000147647**	** *DPYS* **	6.33 × 10^−3^	1.57 × 10^−3^	4.36 × 10^−1^	0.66	0.02	51.90	20.03	88.39	14.90
**ENSG00000117222**	** *RBBP5* **	5.60 × 10^−3^	1.54 × 10^−3^	4.87 × 10^−1^	0.99	0.04	59.64	23.02	87.90	14.82
**ENSG00000058729**	** *RIOK2* **	2.19 × 10^−3^	4.52 × 10^−4^	3.93 × 10^−1^	0.33	0.01	49.47	19.09	87.87	14.81
**ENSG00000023734**	** *STRAP* **	2.31 × 10^−1^	2.20 × 10^−3^	1.61 × 10^−2^	0.33	0.01	2.87	1.11	87.03	14.67
**ENSG00000169242**	** *EFNA1* **	8.49 × 10^−1^	5.64 × 10^−3^	7.12 × 10^−3^	0.67	0.02	1.10	0.42	86.49	14.58
**ENSG00000112679**	** *DUSP22* **	4.83 × 10^−2^	2.23 × 10^−3^	1.01 × 10^−1^	0.66	0.02	14.88	5.75	85.81	14.46
**ENSG00000099341**	** *PSMD8* **	5.49 × 10^−3^	3.34 × 10^−4^	1.53 × 10^−1^	0.33	0.01	24.71	9.54	84.80	14.29
**ENSG00000140718**	** *FTO* **	5.47 × 10^−3^	9.13 × 10^−4^	3.44 × 10^−1^	0.67	0.02	43.32	16.72	84.65	14.27
**ENSG00000154222**	** *CC2D1B* **	1.91 × 10^−2^	9.53 × 10^−4^	1.10 × 10^−1^	0.33	0.01	16.63	6.42	84.48	14.24
**ENSG00000263528**	** *IKBKE* **	2.78 × 10^−3^	2.70 × 10^−4^	1.86 × 10^−1^	0.00	0.00	22.83	8.81	84.07	14.17
**ENSG00000105641**	** *SLC5A5* **	4.13 × 10^−2^	1.42 × 10^−3^	7.12 × 10^−2^	0.33	0.01	10.64	4.11	83.67	14.10
**ENSG00000104237**	** *RP1* **	2.45 × 10^−2^	8.68 × 10^−4^	8.01 × 10^−2^	0.33	0.01	13.22	5.10	83.43	14.06
**ENSG00000143869**	** *GDF7* **	8.15 × 10^−8^	8.37 × 10^−3^	1.04 × 10^−4^	12.26	0.45	3490.31	1347.19	82.10	13.84
**ENSG00000134571**	** *MYBPC3* **	1.01 × 10^−2^	2.02 × 10^−3^	3.69 × 10^−1^	0.66	0.02	40.75	15.73	81.63	13.76
**ENSG00000198646**	** *NCOA6* **	4.46 × 10^−4^	1.60 × 10^−4^	5.54 × 10^−1^	0.00	0.00	60.49	23.35	81.62	13.76
**ENSG00000187244**	** *BCAM* **	4.79 × 10^−3^	8.09 × 10^−4^	2.69 × 10^−1^	0.00	0.00	25.64	9.90	81.53	13.74
**ENSG00000087470**	** *DNM1L* **	8.07 × 10^−3^	1.79 × 10^−3^	4.04 × 10^−1^	0.66	0.02	44.09	17.02	80.20	13.52
**ENSG00000102144**	** *PGK1* **	3.55 × 10^−3^	3.40 × 10^−4^	1.79 × 10^−1^	0.00	0.00	20.53	7.93	79.73	13.44
**ENSG00000067842**	** *ATP2B3* **	1.37 × 10^−2^	1.83 × 10^−3^	2.48 × 10^−1^	0.33	0.01	26.03	10.05	78.92	13.30
**ENSG00000177535**	** *OR2B11* **	1.85 × 10^−1^	5.93 × 10^−3^	4.97 × 10^−2^	0.34	0.01	4.35	1.68	78.84	13.29
**ENSG00000112335**	** *SNX3* **	3.54 × 10^−3^	1.22 × 10^−3^	5.41 × 10^−1^	0.33	0.01	54.64	21.09	77.26	13.02
**ENSG00000134253**	** *TRIM45* **	2.71 × 10^−2^	1.15 × 10^−3^	6.69 × 10^−2^	0.00	0.00	6.41	2.48	77.18	13.01
**ENSG00000178568**	** *ERBB4* **	1.37 × 10^−3^	2.45 × 10^−4^	3.19 × 10^−1^	0.00	0.00	33.44	12.91	77.03	12.98
**ENSG00000168010**	** *ATG16L2* **	5.15 × 10^−3^	1.82 × 10^−3^	5.39 × 10^−1^	0.33	0.01	52.64	20.32	76.83	12.95
**ENSG00000172548**	** *NIPAL4* **	2.05 × 10^−3^	2.75 × 10^−4^	2.49 × 10^−1^	0.00	0.00	26.94	10.40	76.50	12.89
**ENSG00000255587**	** *RAB44* **	4.01 × 10^−2^	8.19 × 10^−4^	3.36 × 10^−2^	0.00	0.00	3.95	1.52	76.45	12.89
**ENSG00000153246**	** *PLA2R1* **	1.52 × 10^−6^	4.52 × 10^−4^	1.75 × 10^−2^	1.98	0.07	1059.16	408.81	76.44	12.88
**ENSG00000055957**	** *ITIH1* **	3.12 × 10^−1^	4.47 × 10^−3^	2.42 × 10^−2^	0.66	0.02	3.59	1.38	76.00	12.81
**ENSG00000121297**	** *TSHZ3* **	4.30 × 10^−3^	3.51 × 10^−4^	1.56 × 10^−1^	0.00	0.00	17.51	6.76	75.15	12.67
**ENSG00000164035**	** *EMCN* **	5.94 × 10^−2^	1.43 × 10^−3^	3.53 × 10^−2^	0.00	0.00	3.10	1.20	75.01	12.64
**ENSG00000261594**	** *TPBGL* **	1.88 × 10^−3^	2.32 × 10^−4^	2.40 × 10^−1^	0.00	0.00	25.97	10.02	74.15	12.50
**ENSG00000171540**	** *OTP* **	3.76 × 10^−1^	1.04 × 10^−2^	4.06 × 10^−2^	0.66	0.02	3.35	1.29	73.67	12.42
**ENSG00000076201**	** *PTPN23* **	1.69 × 10^−1^	2.57 × 10^−3^	3.23 × 10^−2^	0.66	0.02	5.69	2.20	72.42	12.21
**ENSG00000110057**	** *UNC93B1* **	1.57 × 10^−3^	7.44 × 10^−4^	6.78 × 10^−1^	0.33	0.01	66.02	25.48	71.64	12.08
**ENSG00000140522**	** *RLBP1* **	2.46 × 10^−1^	3.95 × 10^−3^	2.60 × 10^−2^	0.33	0.01	2.86	1.11	71.40	12.03
**ENSG00000092054**	** *MYH7* **	1.41 × 10^−4^	6.04 × 10^−4^	4.55 × 10^−1^	0.66	0.02	227.55	87.83	71.32	12.02
**ENSG00000128891**	** *CCDC32* **	1.22 × 10^−4^	3.47 × 10^−4^	5.81 × 10^−1^	0.33	0.01	184.19	71.10	70.18	11.83
**ENSG00000185189**	** *NRBP2* **	4.41 × 10^−3^	5.55 × 10^−4^	2.23 × 10^−1^	0.00	0.00	20.37	7.86	70.17	11.83
**ENSG00000013619**	** *MAMLD1* **	1.10 × 10^−1^	1.38 × 10^−3^	1.79 × 10^−2^	0.00	0.00	1.60	0.62	69.11	11.65
**ENSG00000099899**	** *TRMT2A* **	3.91 × 10^−3^	9.09 × 10^−4^	4.21 × 10^−1^	0.33	0.01	40.08	15.47	68.99	11.63
**ENSG00000149403**	** *GRIK4* **	1.34 × 10^−2^	1.58 × 10^−3^	1.84 × 10^−1^	0.00	0.00	13.42	5.18	68.95	11.62
**ENSG00000188782**	** *CATSPER4* **	4.85 × 10^−2^	1.81 × 10^−3^	5.45 × 10^−2^	0.00	0.00	3.96	1.53	68.65	11.57
**ENSG00000134323**	** *MYCN* **	1.51 × 10^−3^	9.28 × 10^−4^	7.52 × 10^−1^	0.00	0.00	66.32	25.60	68.41	11.53
**ENSG00000121057**	** *AKAP1* **	3.59 × 10^−3^	6.70 × 10^−4^	3.67 × 10^−1^	0.33	0.01	36.33	14.02	68.37	11.52
**ENSG00000156738**	** *MS4A1* **	1.56 × 10^−3^	3.07 × 10^−4^	3.50 × 10^−1^	0.00	0.00	31.86	12.30	67.83	11.43
**ENSG00000074211**	** *PPP2R2C* **	3.95 × 10^−2^	1.78 × 10^−3^	1.33 × 10^−1^	1.68	0.06	21.13	8.16	67.74	11.42
**ENSG00000164292**	** *RHOBTB3* **	6.08 × 10^−3^	7.55 × 10^−4^	2.13 × 10^−1^	0.00	0.00	18.00	6.95	67.69	11.41
**ENSG00000149256**	** *TENM4* **	4.46 × 10^−3^	6.08 × 10^−4^	2.36 × 10^−1^	0.00	0.00	20.35	7.86	67.13	11.31
**ENSG00000092068**	** *SLC7A8* **	1.11 × 10^−2^	7.97 × 10^−4^	1.26 × 10^−1^	0.00	0.00	11.17	4.31	66.75	11.25
**ENSG00000095464**	** *PDE6C* **	1.59 × 10^−3^	5.36 × 10^−4^	5.07 × 10^−1^	0.00	0.00	42.19	16.29	66.73	11.25
**ENSG00000221968**	** *FADS3* **	9.78 × 10^−1^	1.05 × 10^−2^	1.04 × 10^−2^	0.66	0.02	0.83	0.32	66.23	11.16
**ENSG00000153132**	** *CLGN* **	2.12 × 10^−1^	5.37 × 10^−3^	4.11 × 10^−2^	0.33	0.01	3.47	1.34	66.12	11.14
**ENSG00000116661**	** *FBXO2* **	1.95 × 10^−1^	6.13 × 10^−3^	5.03 × 10^−2^	0.33	0.01	3.88	1.50	65.90	11.11
**ENSG00000144589**	** *STK11IP* **	4.25 × 10^−4^	4.28 × 10^−4^	9.98 × 10^−1^	0.00	0.00	95.62	36.91	64.72	10.91
**ENSG00000135114**	** *OASL* **	3.91 × 10^−2^	3.11 × 10^−3^	1.51 × 10^−1^	0.34	0.01	13.34	5.15	63.76	10.75
**ENSG00000186017**	** *ZNF566* **	4.54 × 10^−3^	7.40 × 10^−4^	2.72 × 10^−1^	0.00	0.00	21.42	8.27	63.52	10.71
**ENSG00000148840**	** *PPRC1* **	6.55 × 10^−4^	7.42 × 10^−3^	2.52 × 10^−1^	3.98	0.15	282.54	109.05	63.40	10.68
**ENSG00000248905**	** *FMN1* **	1.45 × 10^−1^	5.57 × 10^−3^	6.47 × 10^−2^	0.34	0.01	5.15	1.99	63.06	10.63
**ENSG00000204428**	** *LY6G5C* **	2.08 × 10^−2^	1.07 × 10^−3^	8.57 × 10^−2^	0.00	0.00	7.10	2.74	62.75	10.58
**ENSG00000176095**	** *IP6K1* **	3.77 × 10^−2^	1.07 × 10^−3^	4.62 × 10^−2^	0.00	0.00	4.11	1.59	61.71	10.40
**ENSG00000113494**	** *PRLR* **	3.92 × 10^−3^	1.68 × 10^−3^	6.25 × 10^−1^	0.33	0.01	50.37	19.44	61.28	10.33
**ENSG00000167302**	** *TEPSIN* **	2.93 × 10^−2^	2.60 × 10^−3^	1.28 × 10^−1^	0.00	0.00	7.33	2.83	60.71	10.23
**ENSG00000007314**	** *SCN4A* **	1.26 × 10^−2^	1.04 × 10^−3^	1.39 × 10^−1^	0.00	0.00	10.53	4.07	60.39	10.18
**ENSG00000160410**	** *SHKBP1* **	8.40 × 10^−3^	1.70 × 10^−3^	3.66 × 10^−1^	0.33	0.01	29.65	11.45	60.35	10.17
**ENSG00000120156**	** *TEK* **	1.77 × 10^−3^	5.46 × 10^−4^	4.77 × 10^−1^	0.00	0.00	36.13	13.95	60.12	10.13
**ENSG00000001167**	** *NFYA* **	3.92 × 10^−3^	7.31 × 10^−4^	3.14 × 10^−1^	0.00	0.00	23.50	9.07	59.90	10.10
**ENSG00000142409**	** *ZNF787* **	2.16 × 10^−3^	8.92 × 10^−4^	5.77 × 10^−1^	0.00	0.00	42.01	16.21	59.45	10.02
**ENSG00000123191**	** *ATP7B* **	4.75 × 10^−4^	5.29 × 10^−4^	9.49 × 10^−1^	0.00	0.00	94.41	36.44	59.29	9.99
**ENSG00000186496**	** *ZNF396* **	3.56 × 10^−2^	3.06 × 10^−3^	1.64 × 10^−1^	0.33	0.01	13.27	5.12	59.06	9.95
**ENSG00000153292**	** *ADGRF1* **	5.98 × 10^−3^	9.03 × 10^−4^	3.11 × 10^−1^	0.33	0.01	27.30	10.54	58.98	9.94
**ENSG00000007216**	** *SLC13A2* **	9.08 × 10^−3^	7.49 × 10^−4^	1.49 × 10^−1^	0.00	0.00	12.14	4.69	58.93	9.93
**ENSG00000168522**	** *FNTA* **	1.09 × 10^−3^	9.17 × 10^−4^	9.22 × 10^−1^	0.33	0.01	76.63	29.58	58.03	9.78
**ENSG00000213672**	** *NCKIPSD* **	8.13 × 10^−3^	8.85 × 10^−4^	1.87 × 10^−1^	0.00	0.00	13.89	5.36	57.98	9.77
**ENSG00000127241**	** *MASP1* **	1.37 × 10^−3^	1.06 × 10^−3^	8.83 × 10^−1^	0.34	0.01	72.33	27.92	57.91	9.76
**ENSG00000284723**	** *OR8S1* **	3.80 × 10^−2^	1.46 × 10^−3^	6.09 × 10^−2^	0.00	0.00	4.41	1.70	57.67	9.72
**ENSG00000183317**	** *EPHA10* **	7.27 × 10^−3^	1.76 × 10^−3^	3.58 × 10^−1^	0.00	0.00	22.47	8.67	57.37	9.67
**ENSG00000198049**	** *AVPR1B* **	1.00 × 10^−2^	1.68 × 10^−3^	2.60 × 10^−1^	0.00	0.00	15.80	6.10	54.89	9.25
**ENSG00000126464**	** *PRR12* **	8.91 × 10^−3^	1.98 × 10^−3^	3.33 × 10^−1^	0.00	0.00	19.25	7.43	53.84	9.07
**ENSG00000198952**	** *SMG5* **	1.06 × 10^−1^	4.30 × 10^−3^	7.41 × 10^−2^	0.33	0.01	5.87	2.27	53.55	9.03
**ENSG00000162676**	** *GFI1* **	3.36 × 10^−2^	2.78 × 10^−3^	1.20 × 10^−1^	0.00	0.00	6.20	2.39	53.52	9.02
**ENSG00000099840**	** *IZUMO4* **	2.10 × 10^−3^	8.95 × 10^−4^	5.94 × 10^−1^	0.00	0.00	39.46	15.23	53.41	9.00
**ENSG00000102935**	** *ZNF423* **	2.04 × 10^−3^	7.68 × 10^−4^	5.48 × 10^−1^	0.00	0.00	36.24	13.99	53.13	8.96
**ENSG00000149654**	** *CDH22* **	2.24 × 10^−3^	6.17 × 10^−4^	4.47 × 10^−1^	0.00	0.00	30.26	11.68	52.98	8.93
**ENSG00000160179**	** *ABCG1* **	1.74 × 10^−5^	2.23 × 10^−4^	1.63 × 10^−1^	0.00	0.00	400.47	154.57	52.95	8.92
**ENSG00000198870**	** *STKLD1* **	6.46 × 10^−3^	9.99 × 10^−4^	2.59 × 10^−1^	0.00	0.00	16.62	6.41	52.27	8.81
**ENSG00000177885**	** *GRB2* **	1.17 × 10^−3^	4.85 × 10^−4^	5.99 × 10^−1^	0.00	0.00	40.48	15.62	52.12	8.78
**ENSG00000144455**	** *SUMF1* **	1.85 × 10^−3^	2.63 × 10^−3^	8.54 × 10^−1^	0.99	0.04	92.62	35.75	51.75	8.72
**ENSG00000095370**	** *SH2D3C* **	4.45 × 10^−4^	5.78 × 10^−4^	8.77 × 10^−1^	0.00	0.00	91.12	35.17	51.36	8.66
**ENSG00000123307**	** *NEUROD4* **	9.84 × 10^−2^	3.75 × 10^−3^	5.09 × 10^−2^	0.00	0.00	2.09	0.81	50.62	8.53
**ENSG00000138363**	** *ATIC* **	1.20 × 10^−1^	2.37 × 10^−3^	4.10 × 10^−2^	0.33	0.01	4.50	1.74	50.51	8.51
**ENSG00000234828**	** *IQCM* **	1.66 × 10^−3^	6.82 × 10^−4^	5.89 × 10^−1^	0.00	0.00	37.86	14.61	50.45	8.50
**ENSG00000167523**	** *SPATA33* **	9.57 × 10^−3^	1.15 × 10^−3^	2.05 × 10^−1^	0.00	0.00	12.62	4.87	50.33	8.48
**ENSG00000142583**	** *SLC2A5* **	1.40 × 10^−4^	5.39 × 10^−4^	4.87 × 10^−1^	0.33	0.01	152.28	58.78	50.31	8.48
**ENSG00000176170**	** *SPHK1* **	1.46 × 10^−2^	2.04 × 10^−3^	2.17 × 10^−1^	0.00	0.00	11.82	4.56	50.26	8.47
**ENSG00000213762**	** *ZNF134* **	1.99 × 10^−1^	3.91 × 10^−3^	2.46 × 10^−2^	0.00	0.00	0.87	0.34	50.15	8.45
**ENSG00000093183**	** *SEC22C* **	1.52 × 10^−2^	2.24 × 10^−3^	2.27 × 10^−1^	0.00	0.00	11.85	4.57	49.62	8.36
**ENSG00000140025**	** *EFCAB11* **	1.84 × 10^−5^	5.57 × 10^−4^	1.25 × 10^−1^	1.00	0.04	302.13	116.61	48.87	8.24
**ENSG00000128335**	** *APOL2* **	4.49 × 10^−2^	3.60 × 10^−3^	1.15 × 10^−1^	0.00	0.00	5.01	1.93	48.84	8.23
**ENSG00000167615**	** *LENG8* **	1.64 × 10^−3^	4.30 × 10^−4^	4.42 × 10^−1^	0.00	0.00	28.48	10.99	48.80	8.23
**ENSG00000279631**	** *AL158211.5* **	5.00 × 10^−2^	3.38 × 10^−3^	9.65 × 10^−2^	0.00	0.00	4.34	1.68	48.71	8.21
**ENSG00000142513**	** *ACP4* **	6.07 × 10^−5^	3.61 × 10^−4^	3.74 × 10^−1^	0.33	0.01	173.85	67.10	48.35	8.15
**ENSG00000129566**	** *TEP1* **	9.30 × 10^−3^	1.49 × 10^−3^	2.58 × 10^−1^	0.00	0.00	14.41	5.56	48.24	8.13
**ENSG00000178999**	** *AURKB* **	1.30 × 10^−1^	6.73 × 10^−4^	8.25 × 10^−3^	0.00	0.00	1.11	0.43	47.82	8.06
**ENSG00000243232**	** *PCDHAC2* **	1.47 × 10^−1^	4.46 × 10^−3^	3.86 × 10^−2^	0.00	0.00	1.33	0.51	47.79	8.06
**ENSG00000158169**	** *FANCC* **	1.29 × 10^−2^	1.22 × 10^−3^	1.63 × 10^−1^	0.00	0.00	9.81	3.79	47.57	8.02
**ENSG00000185359**	** *HGS* **	7.02 × 10^−2^	1.74 × 10^−3^	3.84 × 10^−2^	0.00	0.00	2.41	0.93	47.25	7.96
**ENSG00000167632**	** *TRAPPC9* **	9.22 × 10^−4^	1.91 × 10^−3^	7.03 × 10^−1^	0.67	0.02	103.89	40.10	47.04	7.93
**ENSG00000106511**	** *MEOX2* **	2.97 × 10^−1^	4.32 × 10^−3^	8.45 × 10^−4^	0.33	0.01	0.00	0.00	46.84	7.89
**ENSG00000163075**	** *CFAP221* **	1.81 × 10^−2^	2.53 × 10^−3^	2.12 × 10^−1^	0.00	0.00	10.32	3.98	46.73	7.88
**ENSG00000116329**	** *OPRD1* **	1.11 × 10^−1^	2.69 × 10^−3^	3.37 × 10^−2^	0.00	0.00	1.59	0.61	46.62	7.86
**ENSG00000135414**	** *GDF11* **	1.51 × 10^−2^	1.59 × 10^−3^	1.74 × 10^−1^	0.00	0.00	9.56	3.69	46.53	7.84
**ENSG00000139219**	** *COL2A1* **	2.69 × 10^−2^	2.75 × 10^−3^	1.54 × 10^−1^	0.00	0.00	7.20	2.78	46.47	7.83
**ENSG00000179046**	** *TRIML2* **	1.00 × 10^0^	2.91 × 10^−3^	2.58 × 10^−3^	0.00	0.00	0.00	0.00	46.37	7.82
**ENSG00000174145**	** *NWD2* **	1.88 × 10^−2^	2.00 × 10^−3^	1.68 × 10^−1^	0.00	0.00	8.61	3.32	46.37	7.81
**ENSG00000107338**	** *SHB* **	8.91 × 10^−2^	4.40 × 10^−2^	2.63 × 10^−3^	1.65	0.06	0.00	0.00	46.04	7.76
**ENSG00000172987**	** *HPSE2* **	6.23 × 10^−4^	2.13 × 10^−4^	5.53 × 10^−1^	0.00	0.00	35.00	13.51	45.84	7.73
**ENSG00000136267**	** *DGKB* **	5.27 × 10^−4^	3.19 × 10^−3^	3.56 × 10^−1^	1.03	0.04	178.64	68.95	45.38	7.65
**ENSG00000141577**	** *CEP131* **	1.02 × 10^−1^	4.71 × 10^−3^	6.06 × 10^−2^	0.00	0.00	2.09	0.81	45.37	7.65
**ENSG00000122121**	** *XPNPEP2* **	1.00 × 10^0^	3.44 × 10^−3^	3.05 × 10^−3^	0.00	0.00	0.00	0.00	44.33	7.47
**ENSG00000174871**	** *CNIH2* **	1.81 × 10^−3^	1.02 × 10^−3^	7.24 × 10^−1^	0.00	0.00	41.60	16.06	44.10	7.43
**ENSG00000122176**	** *FMOD* **	5.86 × 10^−4^	6.06 × 10^−4^	9.85 × 10^−1^	0.00	0.00	66.08	25.51	43.89	7.40
**ENSG00000149305**	** *HTR3B* **	1.79 × 10^−2^	1.93 × 10^−3^	1.73 × 10^−1^	0.00	0.00	8.63	3.33	43.83	7.39
**ENSG00000234224**	** *TMEM229A* **	2.93 × 10^−2^	1.41 × 10^−3^	8.34 × 10^−2^	0.00	0.00	4.96	1.92	43.50	7.33
**ENSG00000149269**	** *PAK1* **	1.73 × 10^−3^	1.07 × 10^−3^	7.63 × 10^−1^	0.00	0.00	43.31	16.71	43.24	7.29
**ENSG00000140945**	** *CDH13* **	1.97 × 10^−1^	3.07 × 10^−2^	4.63 × 10^−3^	0.66	0.02	0.00	0.00	43.18	7.28
**ENSG00000111331**	** *OAS3* **	9.44 × 10^−3^	1.99 × 10^−3^	3.26 × 10^−1^	0.00	0.00	15.62	6.03	42.84	7.22
**ENSG00000169783**	** *LINGO1* **	1.08 × 10^−2^	2.01 × 10^−3^	2.92 × 10^−1^	0.00	0.00	14.02	5.41	42.78	7.21
**ENSG00000140470**	** *ADAMTS17* **	2.81 × 10^−2^	2.94 × 10^−3^	1.58 × 10^−1^	0.00	0.00	6.90	2.66	42.70	7.20
**ENSG00000076555**	** *ACACB* **	1.10 × 10^−2^	1.36 × 10^−3^	2.11 × 10^−1^	0.00	0.00	11.00	4.24	42.52	7.17
**ENSG00000104884**	** *ERCC2* **	2.82 × 10^−3^	1.19 × 10^−3^	5.93 × 10^−1^	0.00	0.00	31.44	12.13	42.50	7.16
**ENSG00000112530**	** *PACRG* **	3.89 × 10^−1^	6.94 × 10^−3^	1.90 × 10^−2^	0.00	0.00	0.29	0.11	41.77	7.04
**ENSG00000115694**	** *STK25* **	2.32 × 10^−2^	2.21 × 10^−3^	1.53 × 10^−1^	0.00	0.00	7.21	2.78	41.56	7.00
**ENSG00000102921**	** *N4BP1* **	1.91 × 10^−6^	1.74 × 10^−4^	2.66 × 10^−2^	0.00	0.00	822.43	317.44	41.03	6.91
**ENSG00000169683**	** *LRRC45* **	8.86 × 10^−1^	1.06 × 10^−2^	7.97 × 10^−3^	0.33	0.01	0.30	0.11	40.91	6.90
**ENSG00000140093**	** *SERPINA10* **	2.13 × 10^−1^	6.08 × 10^−3^	3.43 × 10^−2^	0.00	0.00	0.82	0.32	40.74	6.87
**ENSG00000119522**	** *DENND1A* **	3.88 × 10^−3^	1.12 × 10^−3^	4.48 × 10^−1^	0.00	0.00	22.13	8.54	40.41	6.81
**ENSG00000179889**	** *PDXDC1* **	1.68 × 10^−3^	1.58 × 10^−3^	9.70 × 10^−1^	0.33	0.01	56.49	21.81	40.03	6.75
**ENSG00000123977**	** *DAW1* **	3.57 × 10^−4^	2.28 × 10^−4^	8.10 × 10^−1^	0.00	0.00	45.41	17.53	39.92	6.73
**ENSG00000139410**	** *SDSL* **	6.31 × 10^−3^	1.50 × 10^−3^	3.74 × 10^−1^	0.00	0.00	17.55	6.77	39.45	6.65
**ENSG00000100983**	** *GSS* **	3.04 × 10^−1^	1.33 × 10^−2^	2.99 × 10^−3^	0.33	0.01	0.00	0.00	39.25	6.62
**ENSG00000136861**	** *CDK5RAP2* **	2.85 × 10^−1^	5.21 × 10^−4^	3.19 × 10^−3^	0.00	0.00	0.42	0.16	39.15	6.60
**ENSG00000117868**	** *ESYT2* **	1.57 × 10^−2^	1.82 × 10^−3^	1.93 × 10^−1^	0.00	0.00	9.10	3.51	39.10	6.59
**ENSG00000132464**	** *ENAM* **	9.14 × 10^−2^	3.88 × 10^−3^	5.97 × 10^−2^	0.00	0.00	2.15	0.83	39.06	6.58
**ENSG00000119403**	** *PHF19* **	1.91 × 10^−2^	2.36 × 10^−3^	1.98 × 10^−1^	0.00	0.00	8.47	3.27	38.42	6.48
**ENSG00000186352**	** *ANKRD37* **	1.97 × 10^−1^	5.29 × 10^−3^	3.36 × 10^−2^	0.00	0.00	0.87	0.33	38.20	6.44
**ENSG00000110719**	** *TCIRG1* **	4.29 × 10^−2^	3.54 × 10^−3^	1.24 × 10^−1^	0.00	0.00	4.67	1.80	37.83	6.38
**ENSG00000115145**	** *STAM2* **	1.00 × 10^0^	4.00 × 10^−3^	3.55 × 10^−3^	0.00	0.00	0.00	0.00	37.73	6.36
**ENSG00000104419**	** *NDRG1* **	6.84 × 10^−3^	1.79 × 10^−3^	4.02 × 10^−1^	0.00	0.00	17.73	6.84	37.64	6.34
**ENSG00000139990**	** *DCAF5* **	6.90 × 10^−3^	1.66 × 10^−3^	3.78 × 10^−1^	0.00	0.00	16.79	6.48	37.62	6.34
**ENSG00000155096**	** *AZIN1* **	4.73 × 10^−3^	1.60 × 10^−3^	4.98 × 10^−1^	0.00	0.00	22.39	8.64	37.49	6.32
**ENSG00000079999**	** *KEAP1* **	1.19 × 10^−3^	1.85 × 10^−3^	8.12 × 10^−1^	0.33	0.01	71.76	27.70	37.46	6.31
**ENSG00000119285**	** *HEATR1* **	1.80 × 10^−2^	1.14 × 10^−3^	1.20 × 10^−1^	0.00	0.00	6.45	2.49	37.32	6.29
**ENSG00000171954**	** *CYP4F22* **	6.09 × 10^−3^	1.26 × 10^−3^	3.44 × 10^−1^	0.00	0.00	15.54	6.00	36.45	6.14
**ENSG00000140538**	** *NTRK3* **	1.15 × 10^−3^	5.73 × 10^−4^	6.84 × 10^−1^	0.00	0.00	33.31	12.86	36.44	6.14
**ENSG00000036448**	** *MYOM2* **	5.14 × 10^−3^	9.91 × 10^−4^	3.33 × 10^−1^	0.00	0.00	15.70	6.06	36.39	6.13
**ENSG00000177728**	** *TMEM94* **	5.15 × 10^−2^	1.69 × 10^−3^	5.51 × 10^−2^	0.00	0.00	2.92	1.13	36.36	6.13
**ENSG00000188522**	** *FAM83G* **	6.49 × 10^−2^	3.35 × 10^−3^	7.71 × 10^−2^	0.00	0.00	2.91	1.12	36.13	6.09
**ENSG00000189013**	** *KIR2DL4* **	1.72 × 10^−1^	6.32 × 10^−3^	4.59 × 10^−2^	0.00	0.00	1.07	0.41	35.63	6.01
**ENSG00000170085**	** *SIMC1* **	6.09 × 10^−2^	3.29 × 10^−3^	8.18 × 10^−2^	0.00	0.00	3.10	1.20	35.39	5.97
**ENSG00000069431**	** *ABCC9* **	3.51 × 10^−4^	4.76 × 10^−4^	8.64 × 10^−1^	0.00	0.00	63.13	24.37	35.29	5.95
**ENSG00000137055**	** *PLAA* **	2.64 × 10^−4^	1.15 × 10^−3^	4.82 × 10^−1^	0.66	0.02	101.84	39.31	35.26	5.94
**ENSG00000279235**	** *AC011944.2* **	1.00 × 10^0^	9.76 × 10^−3^	8.87 × 10^−3^	0.00	0.00	0.00	0.00	35.21	5.93
**ENSG00000171475**	** *WIPF2* **	1.64 × 10^−3^	1.25 × 10^−3^	8.67 × 10^−1^	0.00	0.00	41.80	16.14	35.09	5.91
**ENSG00000117174**	** *ZNHIT6* **	7.14 × 10^−2^	2.48 × 10^−3^	5.60 × 10^−2^	0.00	0.00	2.42	0.93	34.79	5.86
**ENSG00000180448**	** *ARHGAP45* **	1.90 × 10^−1^	3.25 × 10^−2^	4.49 × 10^−3^	0.67	0.02	0.00	0.00	34.79	5.86
**ENSG00000162415**	** *ZSWIM5* **	8.88 × 10^−1^	2.17 × 10^−2^	1.68 × 10^−2^	0.33	0.01	0.29	0.11	34.35	5.79
**ENSG00000117091**	** *CD48* **	1.00 × 10^0^	9.21 × 10^−3^	8.36 × 10^−3^	0.00	0.00	0.00	0.00	34.21	5.77
**ENSG00000181804**	** *SLC9A9* **	3.83 × 10^−1^	7.48 × 10^−3^	2.11 × 10^−2^	0.00	0.00	0.29	0.11	34.10	5.75
**ENSG00000165238**	** *WNK2* **	2.60 × 10^−2^	2.62 × 10^−3^	1.62 × 10^−1^	0.00	0.00	6.15	2.38	33.80	5.70
**ENSG00000163239**	** *TDRD10* **	2.31 × 10^−3^	7.97 × 10^−4^	5.41 × 10^−1^	0.00	0.00	23.85	9.21	33.70	5.68
**ENSG00000186009**	** *ATP4B* **	1.00 × 10^0^	9.97 × 10^−3^	9.06 × 10^−3^	0.00	0.00	0.00	0.00	33.69	5.68
**ENSG00000169994**	** *MYO7B* **	4.34 × 10^−3^	1.81 × 10^−3^	5.83 × 10^−1^	0.00	0.00	24.04	9.28	33.58	5.66
**ENSG00000162068**	** *NTN3* **	1.00 × 10^0^	9.60 × 10^−3^	8.71 × 10^−3^	0.00	0.00	0.00	0.00	33.16	5.59
**ENSG00000159433**	** *STARD9* **	4.40 × 10^−4^	3.12 × 10^−2^	5.63 × 10^−2^	4.39	0.16	309.78	119.57	32.51	5.48
**ENSG00000187166**	** *H1FNT* **	3.66 × 10^−1^	9.86 × 10^−3^	2.99 × 10^−2^	0.00	0.00	0.33	0.13	32.44	5.47
**ENSG00000167792**	** *NDUFV1* **	3.50 × 10^−2^	1.91 × 10^−3^	9.57 × 10^−2^	0.00	0.00	4.12	1.59	31.52	5.31
**ENSG00000154478**	** *GPR26* **	8.86 × 10^−3^	1.81 × 10^−3^	3.34 × 10^−1^	0.00	0.00	12.61	4.87	31.35	5.28
**ENSG00000106052**	** *TAX1BP1* **	7.96 × 10^−4^	1.12 × 10^−3^	8.39 × 10^−1^	0.00	0.00	58.36	22.52	30.96	5.22
**ENSG00000137210**	** *TMEM14B* **	3.69 × 10^−1^	1.01 × 10^−2^	3.06 × 10^−2^	0.00	0.00	0.33	0.13	30.66	5.17
**ENSG00000008735**	** *MAPK8IP2* **	4.07 × 10^−3^	1.81 × 10^−3^	6.16 × 10^−1^	0.00	0.00	23.36	9.02	30.37	5.12
**ENSG00000173614**	** *NMNAT1* **	3.16 × 10^−1^	3.21 × 10^−2^	8.21 × 10^−3^	0.33	0.01	0.00	0.00	30.14	5.08
**ENSG00000132000**	** *PODNL1* **	3.80 × 10^−6^	1.01 × 10^−4^	1.12 × 10^−1^	0.00	0.00	220.70	85.19	29.82	5.03
**ENSG00000198223**	** *CSF2RA* **	3.12 × 10^−1^	3.12 × 10^−2^	7.89 × 10^−3^	0.33	0.01	0.00	0.00	29.75	5.01
**ENSG00000112159**	** *MDN1* **	4.47 × 10^−4^	4.12 × 10^−3^	2.80 × 10^−1^	0.99	0.04	130.24	50.27	29.58	4.99
**ENSG00000123080**	** *CDKN2C* **	1.27 × 10^−1^	6.29 × 10^−3^	6.70 × 10^−2^	0.00	0.00	1.59	0.61	28.83	4.86
**ENSG00000183840**	** *GPR39* **	1.61 × 10^−1^	4.81 × 10^−3^	4.04 × 10^−2^	0.00	0.00	1.09	0.42	28.69	4.84
**ENSG00000197312**	** *DDI2* **	9.52 × 10^−3^	2.02 × 10^−3^	3.46 × 10^−1^	0.00	0.00	11.81	4.56	28.44	4.79
**ENSG00000137504**	** *CREBZF* **	2.29 × 10^−1^	4.41 × 10^−3^	2.56 × 10^−2^	0.00	0.00	0.65	0.25	28.38	4.78
**ENSG00000177352**	** *CCDC71* **	1.00 × 10^0^	1.20 × 10^−2^	1.10 × 10^−2^	0.00	0.00	0.00	0.00	28.34	4.78
**ENSG00000169635**	** *HIC2* **	1.38 × 10^−5^	2.71 × 10^−4^	1.32 × 10^−1^	0.00	0.00	210.03	81.07	27.91	4.70
**ENSG00000103365**	** *GGA2* **	1.38 × 10^−1^	5.74 × 10^−3^	5.63 × 10^−2^	0.00	0.00	1.34	0.52	27.64	4.66
**ENSG00000104140**	** *RHOV* **	1.00 × 10^0^	9.51 × 10^−3^	8.57 × 10^−3^	0.00	0.00	0.00	0.00	27.27	4.60
**ENSG00000167822**	** *OR8J3* **	1.03 × 10^−3^	2.44 × 10^−2^	1.05 × 10^−1^	1.65	0.06	280.21	108.15	27.17	4.58
**ENSG00000114650**	** *SCAP* **	3.02 × 10^−2^	2.72 × 10^−3^	1.57 × 10^−1^	0.00	0.00	5.08	1.96	27.06	4.56
**ENSG00000142552**	** *RCN3* **	1.00 × 10^0^	1.27 × 10^−2^	1.15 × 10^−2^	0.00	0.00	0.00	0.00	26.65	4.49
**ENSG00000131061**	** *ZNF341* **	2.31 × 10^−4^	7.85 × 10^−4^	4.92 × 10^−1^	0.00	0.00	86.09	33.23	26.65	4.49
**ENSG00000123388**	** *HOXC11* **	1.00 × 10^0^	1.16 × 10^−2^	1.05 × 10^−2^	0.00	0.00	0.00	0.00	26.49	4.46
**ENSG00000173334**	** *TRIB1* **	3.20 × 10^−1^	9.57 × 10^−3^	3.95 × 10^−2^	0.00	0.00	0.46	0.18	26.40	4.45
**ENSG00000095906**	** *NUBP2* **	1.04 × 10^−3^	1.26 × 10^−3^	9.13 × 10^−1^	0.00	0.00	44.23	17.07	26.36	4.44
**ENSG00000204385**	** *SLC44A4* **	1.46 × 10^−3^	3.19 × 10^−3^	6.12 × 10^−1^	0.00	0.00	77.60	29.95	26.33	4.44
**ENSG00000283654**	** *LMLN2* **	9.53 × 10^−4^	3.67 × 10^−3^	4.64 × 10^−1^	0.33	0.01	88.90	34.31	26.31	4.43
**ENSG00000059145**	** *UNKL* **	5.72 × 10^−2^	3.49 × 10^−3^	9.76 × 10^−2^	0.00	0.00	2.91	1.12	25.41	4.28
**ENSG00000165061**	** *ZMAT4* **	1.00 × 10^0^	1.31 × 10^−2^	1.19 × 10^−2^	0.00	0.00	0.00	0.00	25.21	4.25
**ENSG00000186106**	** *ANKRD46* **	9.98 × 10^−4^	1.21 × 10^−3^	9.11 × 10^−1^	0.00	0.00	42.14	16.27	25.04	4.22
**ENSG00000278619**	** *MRM1* **	1.00 × 10^0^	1.04 × 10^−2^	9.35 × 10^−3^	0.00	0.00	0.00	0.00	24.69	4.16
**ENSG00000103995**	** *CEP152* **	1.10 × 10^−1^	5.16 × 10^−3^	7.06 × 10^−2^	0.00	0.00	1.69	0.65	24.50	4.13
**ENSG00000165071**	** *TMEM71* **	1.00 × 10^0^	1.07 × 10^−2^	9.69 × 10^−3^	0.00	0.00	0.00	0.00	24.49	4.13
**ENSG00000131023**	** *LATS1* **	1.06 × 10^−1^	5.34 × 10^−3^	7.37 × 10^−2^	0.00	0.00	1.75	0.68	24.26	4.09
**ENSG00000130413**	** *STK33* **	3.68 × 10^−1^	1.08 × 10^−2^	3.34 × 10^−2^	0.00	0.00	0.33	0.13	23.91	4.03
**ENSG00000196967**	** *ZNF585A* **	2.26 × 10^−3^	2.09 × 10^−3^	9.61 × 10^−1^	0.00	0.00	33.07	12.76	23.90	4.03
**ENSG00000138162**	** *TACC2* **	2.26 × 10^−1^	8.83 × 10^−3^	5.26 × 10^−2^	0.00	0.00	0.76	0.29	23.81	4.01
**ENSG00000140015**	** *KCNH5* **	1.00 × 10^0^	7.39 × 10^−3^	6.57 × 10^−3^	0.00	0.00	0.00	0.00	23.68	3.99
**ENSG00000177600**	** *RPLP2* **	1.16 × 10^−3^	1.20 × 10^−3^	9.89 × 10^−1^	0.00	0.00	35.41	13.67	23.66	3.99
**ENSG00000149923**	** *PPP4C* **	1.00 × 10^0^	5.82 × 10^−3^	5.14 × 10^−3^	0.00	0.00	0.00	0.00	23.56	3.97
**ENSG00000198104**	** *OR2T6* **	1.11 × 10^−2^	9.05 × 10^−4^	1.76 × 10^−1^	0.00	0.00	6.76	2.61	23.44	3.95
**ENSG00000159882**	** *ZNF230* **	3.85 × 10^−1^	1.30 × 10^−2^	3.62 × 10^−2^	0.00	0.00	0.29	0.11	23.25	3.92
**ENSG00000085719**	** *CPNE3* **	1.00 × 10^0^	7.53 × 10^−3^	6.70 × 10^−3^	0.00	0.00	0.00	0.00	22.48	3.79
**ENSG00000183921**	** *SDR42E2* **	1.00 × 10^0^	1.45 × 10^−2^	1.32 × 10^−2^	0.00	0.00	0.00	0.00	22.07	3.72
**ENSG00000120910**	** *PPP3CC* **	1.00 × 10^0^	1.58 × 10^−2^	1.44 × 10^−2^	0.00	0.00	0.00	0.00	21.89	3.69
**ENSG00000169813**	** *HNRNPF* **	1.09 × 10^−1^	4.82 × 10^−3^	6.61 × 10^−2^	0.00	0.00	1.60	0.62	21.86	3.68
**ENSG00000169976**	** *SF3B5* **	5.33 × 10^−4^	1.90 × 10^−3^	4.54 × 10^−1^	0.00	0.00	80.95	31.24	21.75	3.67
**ENSG00000205517**	** *RGL3* **	1.00 × 10^0^	1.49 × 10^−2^	1.36 × 10^−2^	0.00	0.00	0.00	0.00	21.73	3.66
**ENSG00000149136**	** *SSRP1* **	6.70 × 10^−3^	1.13 × 10^−3^	3.22 × 10^−1^	0.00	0.00	9.78	3.77	21.66	3.65
**ENSG00000108684**	** *ASIC2* **	1.00 × 10^0^	1.45 × 10^−2^	1.32 × 10^−2^	0.00	0.00	0.00	0.00	21.61	3.64
**ENSG00000196352**	** *CD55* **	1.67 × 10^−3^	8.10 × 10^−4^	6.85 × 10^−1^	0.00	0.00	20.16	7.78	21.55	3.63
**ENSG00000123415**	** *SMUG1* **	3.74 × 10^−1^	7.74 × 10^−3^	2.33 × 10^−2^	0.00	0.00	0.29	0.11	21.53	3.63
**ENSG00000144040**	** *SFXN5* **	1.00 × 10^0^	1.33 × 10^−2^	1.21 × 10^−2^	0.00	0.00	0.00	0.00	21.45	3.62
**ENSG00000119760**	** *SUPT7L* **	5.17 × 10^−4^	2.65 × 10^−3^	3.28 × 10^−1^	0.00	0.00	110.83	42.78	21.01	3.54
**ENSG00000141219**	** *C17orf80* **	1.00 × 10^0^	1.29 × 10^−2^	1.17 × 10^−2^	0.00	0.00	0.00	0.00	19.88	3.35
**ENSG00000274209**	** *ANTXRL* **	1.00 × 10^0^	1.64 × 10^−2^	1.50 × 10^−2^	0.00	0.00	0.00	0.00	19.80	3.34
**ENSG00000164535**	** *DAGLB* **	1.00 × 10^0^	1.66 × 10^−2^	1.51 × 10^−2^	0.00	0.00	0.00	0.00	19.66	3.31
**ENSG00000169249**	** *ZRSR2* **	1.00 × 10^0^	1.65 × 10^−2^	1.50 × 10^−2^	0.00	0.00	0.00	0.00	19.44	3.28
**ENSG00000187800**	** *PEAR1* **	1.00 × 10^0^	1.67 × 10^−2^	1.53 × 10^−2^	0.00	0.00	0.00	0.00	19.41	3.27
**ENSG00000213221**	** *DNLZ* **	1.00 × 10^0^	1.58 × 10^−2^	1.44 × 10^−2^	0.00	0.00	0.00	0.00	19.29	3.25
**ENSG00000160951**	** *PTGER1* **	1.14 × 10^−3^	2.69 × 10^−3^	6.02 × 10^−1^	0.00	0.00	55.05	21.25	19.26	3.25
**ENSG00000132517**	** *SLC52A1* **	1.00 × 10^0^	1.71 × 10^−2^	1.56 × 10^−2^	0.00	0.00	0.00	0.00	18.70	3.15
**ENSG00000008300**	** *CELSR3* **	1.54 × 10^−3^	1.65 × 10^−3^	9.71 × 10^−1^	0.00	0.00	28.66	11.06	18.67	3.15
**ENSG00000128052**	** *KDR* **	8.91 × 10^−4^	3.73 × 10^−3^	3.78 × 10^−1^	0.00	0.00	89.21	34.43	18.65	3.14
**ENSG00000183307**	** *TMEM121B* **	3.21 × 10^−4^	1.42 × 10^−3^	4.12 × 10^−1^	0.00	0.00	70.06	27.04	18.43	3.11
**ENSG00000163687**	** *DNASE1L3* **	1.00 × 10^0^	1.79 × 10^−2^	1.63 × 10^−2^	0.00	0.00	0.00	0.00	18.25	3.08
**ENSG00000169679**	** *BUB1* **	1.00 × 10^0^	6.44 × 10^−3^	5.63 × 10^−3^	0.00	0.00	0.00	0.00	17.66	2.98
**ENSG00000164488**	** *DACT2* **	3.76 × 10^−4^	7.60 × 10^−3^	1.46 × 10^−1^	0.66	0.02	121.59	46.93	17.57	2.96
**ENSG00000151952**	** *TMEM132D* **	1.36 × 10^−3^	3.08 × 10^−3^	6.24 × 10^−1^	0.00	0.00	47.33	18.27	17.34	2.92
**ENSG00000010292**	** *NCAPD2* **	1.00 × 10^0^	3.21 × 10^−3^	2.74 × 10^−3^	0.00	0.00	0.00	0.00	17.33	2.92
**ENSG00000164076**	** *CAMKV* **	1.00 × 10^0^	1.89 × 10^−2^	1.72 × 10^−2^	0.00	0.00	0.00	0.00	17.13	2.89
**ENSG00000205978**	** *NYNRIN* **	7.33 × 10^−7^	4.92 × 10^−3^	8.47 × 10^−4^	1.00	0.04	751.28	289.98	16.86	2.84
**ENSG00000104881**	** *PPP1R13L* **	1.96 × 10^−1^	7.19 × 10^−3^	5.01 × 10^−2^	0.00	0.00	0.83	0.32	16.59	2.80
**ENSG00000280789**	** *PAGR1* **	1.00 × 10^0^	1.80 × 10^−2^	1.64 × 10^−2^	0.00	0.00	0.00	0.00	16.57	2.79
**ENSG00000167702**	** *KIFC2* **	1.02 × 10^−3^	2.57 × 10^−3^	5.96 × 10^−1^	0.00	0.00	44.49	17.17	15.92	2.68
**ENSG00000104951**	** *IL4I1* **	1.00 × 10^0^	1.93 × 10^−2^	1.76 × 10^−2^	0.00	0.00	0.00	0.00	15.90	2.68
**ENSG00000167617**	** *CDC42EP5* **	1.00 × 10^0^	1.99 × 10^−2^	1.82 × 10^−2^	0.00	0.00	0.00	0.00	15.89	2.68
**ENSG00000168675**	** *LDLRAD4* **	2.01 × 10^−3^	1.33 × 10^−2^	2.94 × 10^−1^	0.34	0.01	85.19	32.88	15.78	2.66
**ENSG00000167513**	** *CDT1* **	1.52 × 10^−1^	6.36 × 10^−3^	6.60 × 10^−2^	0.00	0.00	1.09	0.42	15.59	2.63
**ENSG00000166682**	** *TMPRSS5* **	1.00 × 10^0^	2.28 × 10^−2^	2.08 × 10^−2^	0.00	0.00	0.00	0.00	14.52	2.45
**ENSG00000157450**	** *RNF111* **	1.91 × 10^−4^	5.83 × 10^−3^	9.49 × 10^−2^	0.33	0.01	138.09	53.30	14.51	2.45
**ENSG00000169085**	** *VXN* **	1.77 × 10^−3^	1.39 × 10^−2^	2.56 × 10^−1^	0.33	0.01	85.31	32.93	14.38	2.42
**ENSG00000172508**	** *CARNS1* **	1.00 × 10^0^	2.28 × 10^−2^	2.08 × 10^−2^	0.00	0.00	0.00	0.00	14.16	2.39
**ENSG00000112115**	** *IL17A* **	1.47 × 10^−1^	5.73 × 10^−3^	5.97 × 10^−2^	0.00	0.00	1.10	0.42	14.13	2.38
**ENSG00000188747**	** *NOXA1* **	1.00 × 10^0^	2.17 × 10^−2^	1.98 × 10^−2^	0.00	0.00	0.00	0.00	14.05	2.37
**ENSG00000145907**	** *G3BP1* **	8.05 × 10^−4^	2.95 × 10^−3^	4.54 × 10^−1^	0.00	0.00	50.47	19.48	13.81	2.33
**ENSG00000135362**	** *PRR5L* **	1.00 × 10^0^	2.13 × 10^−2^	1.94 × 10^−2^	0.00	0.00	0.00	0.00	13.45	2.27
**ENSG00000048342**	** *CC2D2A* **	1.00 × 10^0^	2.14 × 10^−2^	1.95 × 10^−2^	0.00	0.00	0.00	0.00	13.15	2.22
**ENSG00000161547**	** *SRSF2* **	1.00 × 10^0^	5.18 × 10^−3^	4.43 × 10^−3^	0.00	0.00	0.00	0.00	12.98	2.19
**ENSG00000173611**	** *SCAI* **	1.00 × 10^0^	2.12 × 10^−2^	1.93 × 10^−2^	0.00	0.00	0.00	0.00	12.59	2.12
**ENSG00000101447**	** *FAM83D* **	1.00 × 10^0^	2.10 × 10^−2^	1.90 × 10^−2^	0.00	0.00	0.00	0.00	12.37	2.09
**ENSG00000124228**	** *DDX27* **	1.18 × 10^−1^	4.36 × 10^−3^	6.40 × 10^−2^	0.00	0.00	1.15	0.44	11.05	1.86
**ENSG00000164405**	** *UQCRQ* **	1.00 × 10^0^	2.29 × 10^−2^	2.08 × 10^−2^	0.00	0.00	0.00	0.00	10.97	1.85
**ENSG00000103423**	** *DNAJA3* **	1.00 × 10^0^	1.54 × 10^−2^	1.36 × 10^−2^	0.00	0.00	0.00	0.00	10.70	1.80

**Table 4 genes-14-00853-t004:** The novel (unannotated) transcripts that were significantly different between the cells, and the EVs chromosomal coordinates.

Significantly Enriched in EVs			
Chr	Start	End	Name
20	37437053	37437078	novel475_26bp
2	185057197	185057222	novel410_26bp
12	129972983	129973001	novel153_19bp
X	117500158	117500185	novel914_28bp
13	28624284	28624304	novel156_21bp
8	114852333	114852352	novel808_20bp
Significantly higher in cells			
Chr	start	end	name
7	145997340	145997428	novel794_89bp
6	27177193	27177214	novel692_22bp
20	48612475	48612492	novel478_18bp
MT	311	408	novel894_98bp
11	10510171	10510209	novel98_39bp
11	10509189	10509302	novel92_114bp

## Data Availability

The RNA-seq data have been deposited in the European Nucleotide Archive (EMBL-EBI) under accession number: PRJEB52845.

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
