# Peer review of "Human Adult Astrocyte Extracellular Vesicle Transcriptomics Study Identifies Specific RNAs Which Are Preferentially Secreted as EV Luminal Cargo"

_genes, 2023, doi:10.3390/genes14040853_

Round 1

Reviewer 1 Report

This study was designed to characterize secreted extracellular vesicles (EVs), and RNA cargo of human astrocytes derived from adult brain, and found that CD81 is the dominant tetraspanin of human astrocytes-derived EVs while integrin β1 is the characteristic of the larger EVs. Furthermore, the comparison of cellular RNAs, EVs-derived RNAs and RNAs in ProteinaseK/RNase-treated EVs identified secreted protected RNAs and some unannotated RNAs in EVs. This reviewer has some major concerns about this manuscript:

1.     Compared with human astrocytes-derived intact EVs, what is the functional specificity of ProteinaseK/RNase-treated EVs? What are the respective roles of ProteinaseK/RNase-digested and -undigested RNAs? It will be meaningful to provide some experimental evidence.

2.     What is the mechanism underlying the protection of special secreted RNAs against ProteinaseK/RNase digestion?

3.     The manuscript is not well written, and the organization of the manuscript should be improved. For example, the structure of the Introduction is fragmented.

Author Response

1. Compared with human astrocytes-derived intact EVs, what is the functional specificity of ProteinaseK/RNase-treated EVs? What are the respective roles of ProteinaseK/RNase-digested and -undigested RNAs? It will be meaningful to provide some experimental evidence.

Answer to the first part of question 1: EV surface is negatively charged and attracts protein and nucleic acids to form a corona. This corona is not likely to be completely identical if the EVs are collected from cell culture or in the brain extracellular fluid, but the luminal cargo does not include cell culture medium component and should be more representative. For this reason protocols were developed to distinguish RNA which is inside the EVs lumen from RNA stuck to the surface and their use is recommended for EV characterization work in MISEV2018.

Our answer to the second part of the question is :

Even though the majority of extracellular RNA in body fluids is not in EVs, their functions are not well studied, in our manuscript we cited a manuscript describing the EV-mediated transfer of miR21 from cell to cell (Abels et al. 2019, Cell Rep. 2019, 28, 3105-3119.e7). To illustrate that non-EV miRNA may have functions we added a sentence about the identified function of non-EV miRNA (line 556-557), citing Park et al. 2014 showing that extracellular Let-7b activates cell surface receptor: TLR7 (Neuron 82, 47-54) of sensory neurons eliciting pain and the study of Han et al. 2018 (Neuron 99(3)449-463) describing the specific binding of miR-711 on TRPA1 channels at the cell surface and its effect on itching in mouse. A part of the RNA and EVs found in body fluids may also represent waste product from the cells.

2.  What is the mechanism underlying the protection of special secreted RNAs against ProteinaseK/RNase digestion?

EVs have a double lipidic membrane similar to cellular membranes, which proteinase K and RNAse cannot enter, RNA residing in EV lumen is therefore protected unlike RNA in protein aggregates for instance

3. The manuscript is not well written, and the organization of the manuscript should be improved. For example, the structure of the Introduction is fragmented. 

We have reorganized the introduction to improve the readability of the manuscript

Reviewer 2 Report

The recommendations for improvement are listed below:

1. As the authors claimed, some RNAs are preferentially secreted as EV luminal cargo. They set up a proteinase K plus RNase protection assay to get this conclusion. Based on the EV isolation method, it must contaminate many RNPs and protein aggregates. This conclusion would be more convincing if triton, proteinase K and RNase were added to the experimental group.

2. It is a weakness and concern that the authors reported the results for cells derived just from 1 donor. As we know, individuals vary greatly in biological studies. The source of samples is very limited, which is understandable, but the broad spectrum of RNA analysis from a single donor is questionable.

3. For figure 1 and figure 2, the author used “ NO_FBSvsK”, it would be more clear to describe what K stands for in the figure legends.

Author Response

As the authors claimed, some RNAs are preferentially secreted as EV luminal cargo. They set up a proteinase K plus RNase protection assay to get this conclusion. Based on the EV isolation method, it must contaminate many RNPs and protein aggregates. This conclusion would be more convincing if triton, proteinase K and RNase were added to the experimental group

reply:  Adding triton destroys the EVs completely and leaves the protein aggregates that protect RNA intact: triton and RNAse treatment can be done to remove EV RNA and look for lost RNA in the data, which is the opposite of what we have done here, but of course a completely valid alternative. 

But adding triton, proteinase k and RNAse would result in removing all RNA from the preparation without indicating whether it was in EV lumen or in protein aggregates.

2. It is a weakness and concern that the authors reported the results for cells derived just from 1 donor. As we know, individuals vary greatly in biological studies. The source of samples is very limited, which is understandable, but the broad spectrum of RNA analysis from a single donor is questionable.

Unfortunately there are few commercially available human astrocytes and most of them are fetal.

There are a few mouse and rat studies, usually made with astrocytes collected from several one day old pups (mixed together), which report astrocyte EV miRNA.

We made QPCR with RNA from 1 new astrocyte cell batch derived from a 58 year old male we bought from CellApplication Inc, San Diego, USA (cat Ne 882A-05). We only have one batch of RNA from each condition. The miRNA tested were all detected in cells with Ct values between 23 (miR-let7b-5p) and 32 (miR-148a-3p) and EVs, after normalizing for amount of cDNA the trend is similar as what we observed in miRNA-seq data, as there are no replicates and these are preliminary results, we did not include them in the main manuscript but they are presented in the supplementary files as Figure S1.

For figure 1 and figure 2, the author used “ NO_FBSvsK”, it would be more clear to describe what K stands for in the figure legends.

We made the suggested addition to both figures's legend

Reviewer 3 Report

Keerthanaa Balasubramanian Shanthi et al compared the cellular microRNA content to the matched EV microRNAs of primary astrocyte cultured in serum-free condition. The results may provide a basis and reference for the study of neuronal regulation. However, some parts of the manuscript were challenging to understand.

1. I would recommend additional editing of the figures (combine figure 7 and table 1 into one figure, as well as figure 8 and table 2) to improve clarity for the reader, as they are in their current format unnecessarily large and redundant.

2. the limit of this study as the authors said is the sample size just 1. The website of innoprot showed other primary astrocytes they sell?

3. the validation of mRNA/microRNA by qRT-PCR is lack

Author Response

 1. I would recommend additional editing of the figures (combine figure 7 and table 1 into one figure, as well as figure 8 and table 2) to improve clarity for the reader, as they are in their current format unnecessarily large and redundant.

Answer: In final online manuscript tables can be open in another window so it is not a big problem, but we are aware that for reviewers tables are not presented in an easy way to read. Combining figure 7 and table 1 would make a crowded figure, so we suggest keeping them as they are, but we agree that table 2 is too big and we will cut it to 34 entries in the main text, the complete table will be submitted with the supplementary files.  Like for figure 7, combining table and figure would make it less readable.

2. the limit of this study as the authors said is the sample size just 1. The website of innoprot showed other primary astrocytes they sell?

We have not been able to get more primary astrocyte from adult donor from innoprot.

We made QPCR with RNA from 1 new astrocyte cell batch derived from a 58 year old male we bought from CellApplication Inc, San Diego, USA (cat Ne 882A-05). We only have one batch from each condition. The miRNA tested were all detected in cells with Ct values between 23 (miR-let7b-5p) and 32 (miR-148a-3p) and EVs, after normalizing for amount of cDNA the trend is similar as what we observed in miRNA-seq data, as there are no replicates, we did not include it in the main manuscript but it can be found in the supplementary files as Figure S1.

3. the validation of mRNA/microRNA by qRT-PCR is lack

As we stated in our answer to question 2 we did test  QPCR with some miRNA that we had found in the study.

Round 2

Reviewer 1 Report

1.    The authors have addressed my concerns and changed the manuscript appropriately. The only remaining issue I have is the English expression. For example, “As a result our protein/particle ratio is 1.84 X 108 which is about 100- 650 fold higher than pure EVs as a mitigating measure we used Proteinase K and RNase A/T1 treatment of EV-enriched fractions to eliminate protein and RNA sticking to the EV surface as well as ribonucleoprotein, and we based our analysis on miRNA which were significantly higher in treated EVs than cells (Line 650-654)”. The sentence should be better broken.

Author Response

We agree that this sentence should be better formulated, we have changed for the following in the text:

As a result our protein/particle ratio is 1.84 X 108, which is about 100 fold higher than the values expected for pure EVs [94]. To eliminate the contaminating proteins and RNA sticking to the EV surface, we used Proteinase K and RNase A/T1 treatment of EV-enriched fractions. Downstream analysis were done with miRNAs, which were significantly higher in treated EVs than cells.

Reviewer 2 Report

As authors mentioned, adding triton, proteinase k and RNAse would result in removing all RNA from the preparation without indicating whether it was in EV lumen or in protein aggregates.  However, the author already claimed some RNAs were in the EV lumen. The author can test these RNAs are sensitive RNase plus Triton or not.  

Author Response

We now tested whether adding 1% triton X100 to the proteinase K/RNase digestion has an effect on the detection of miRNA compared to ProteinaseK /RNAse alone.

We measured DNA concentration after the cDNA synthesis and used the same amount of DNA for EVs, EV-Proteinase K/ RNAse and ProteinaseK/tritonX100/RNAse for let7b-5p and miR24-3p the ct value increased to 35, for miR99a-5p to 37, from 27 (let7b-5p),  25 (miR24-3p) and 29 (miR99a-5p) in EV-proteinaseK/RNAse.

We added the following sentence to the text page 16, lane 402:

Performing ProteinaseK/ RNaseA/T1 treatment in presence of 1% triton X100 resulted in reduction to undetectable level of tested miRNA (ct values above 35 for hsa-let-7b-5p, hsa-miR-24-3p and hsa-miR-99a-5p).